



# Critical soil moisture detection and water-energy limit shift attribution using satellite-based water and carbon fluxes over China

Yi Liu[1], Jingfeng Xiao[2], Xing Li[3], Yue Li[4]

[1]School of Civil Engineering and Architecture, Guangxi University, Nanning 530004, China
[2]Earth Systems Research Center, Institute for the Study of Earth, Oceans, and Space, University of New Hampshire, Durham, NH 03824, USA
[3]School of Geography and Planning, Sun Yat-Sen University, Guangzhou 510275, China
[4]Department of Earth Sciences, Indiana University-Purdue University Indianapolis (IUPUI), Indianapolis, IN 46202, USA

*Correspondence to*: Yi Liu (liuyi.15b@igsnrr.ac.cn)

**Abstract.** Critical soil moisture (CSM), a tipping point of soil moisture (SM) when evapotranspiration (ET) begins to suffer from water limitation, is essential for the vegetation state and corresponding land-atmosphere coupling. The water and energy-limited regime in various biomes and climates shifts under global climate change. However, detecting CSM and attributing water-energy limit shifts to climate and ecosystem variables are challenging as in-situ observations of water, carbon fluxes, and SM are sparse. In this study, CSM derived from two satellite-based methods were assessed over China: the difference between the correlation between SM and ET and the correlation between vapor pressure deficit (VPD) and ET (correlation-difference method) using four satellite-based ET products; the covariance between VPD and gross primary production (GPP) (VPD-GPP-SM method) using four satellite-based GPP products. The ET and GPP products were Penman-Monteith-Leuning (PML) ET and GPP, Global LAnd Surface Satellite (GLASS) ET and GPP, Collocation-Analyzed Multi-source Ensembled Land Evapotranspiration (CAMELE) ET, Surface Energy Balance Algorithm for Land evapotranspiration in China (SEBAL) ET, Two-Leaf light use efficiency model based (TL) GPP, and SIF-based GPP (GOSIF). At flux sites, satellite-based ET and GPP were evaluated by the eddy covariance technique, and CSMs derived from the site-level GPP and ET data were evaluated by the evaporative fraction-soil moisture (EF-SM) methods. Our study revealed that the performance of ET, GPP, and CSM at the site scale demonstrated reliable results and applicability to regional scales. The intercomparison of CSM from multi-source ET and GPP datasets across China indicated their consistency and robustness. Generally, CSM decreased from southern to northern regions of China and decreased with increasing layer depth, particularly in the Tarim Basin and Haihe River Basin. Areas characterized by clay-rich soils (e.g., 0.39 $m^3/m^3$ using GOSIF GPP and 10 cm depth SM), and forests (e.g., 0.37 $m^3/m^3$ using GOSIF GPP and 10 cm depth SM), and located within the Pearl River Basin and Southeastern River Basin displayed a relatively high CSM. Decreased energy limitations in western and southern regions in June–September over the period 2001–2018 were associated with increasing ET and decreasing precipitation, respectively. The findings highlight the variability in CSM and its primary determinants, offering valuable insights into the potential water limitation on both ET and GPP processes under comparable SM circumstances.





# 1 Introduction

Critical soil moisture (CSM), a tipping point of soil moisture (SM) when evapotranspiration (ET) begins to suffer from water limitation, serves as an indicator of shifts in the relationship between water and energy availability (Schwingshackl et al., 2017; Denissen et al., 2020). It is essential in shaping regional climates and indicates the interplay and disconnection of water, energy, and carbon cycles. Below this threshold, ET is primarily influenced by the SM availability in conditions of restricted water supply. Over extended temporal periods, this phenomenon may lead to the persistence of arid and high-temperature conditions (Zhang et al., 2020). Under energy-limited regimes, ET is primarily influenced by the availability of energy, namely temperature and radiation (Rodriguez-Iturbe, 2000; Seneviratne et al., 2010). Water and energy limit shifts may be further strengthened by the interaction between the land and atmosphere, particularly when positive feedback mechanisms such as evaporative cooling exist under energy-limited regimes (Seneviratne et al., 2010). Consequently, it is needed to quantify the characteristics of CSM and its influencing environmental factors.

There has been a customary association between certain CSM values and typical matric potentials for distinct flora and soil types (Homaee et al., 2002). This association helps to define water and energy limit shifts across various biomes and climatic zones. Plants regulate their stomatal resistance to adapt to variations in water stress and atmospheric energy availability, specifically in relation to leaf temperature. This adjustment leads to a decrease in both the transpiration process as the largest part of ET (Good et al., 2015) and the photosynthetic activity reflected by gross primary production (GPP) when exposed to warm and dry air above the canopy (Grossiord et al., 2020; Li, X. et al., 2023). A decrease in ET has been found to result in elevated surface temperature and vapor pressure deficit (VPD) (Gentine et al., 2019). This, in turn, leads to increased atmospheric aridity on a large spatial scale, thereby intensifying the depletion of SM. Consequently, positive land-atmosphere feedback known as the "dry gets dryer" effect occurs (Seneviratne et al., 2010; Gentine et al., 2019). To that end, previous studies have examined climate-SM feedback using different metrics and both observation and simulation data (Seneviratne et al., 2006; Koster et al., 2009; Teuling et al., 2009). According to Rodriguez-Iturbe (2000), a decrease in SM below the CSM results in a decrease in latent heat flux (LE), while sensible heat flux (H) increases. SM thereby exhibits a correlation with the ratio of LE to the total of LE and H, known as evaporative fraction (EF). Laio et al. (2001) and Porporato et al. (2002) found that the EF-SM relationship follows a linear trend during this period. Above CSM, there is no further alteration in EF with SM increases (Feldman et al., 2019). Thus, CSM can be identified when SM exceeds the critical value via the EF-SM framework (Rodriguez-Iturbe, 2000; Seneviratne et al., 2010; Akbar et al., 2018). Currently, computational models (Schwingshackl et al., 2017) and field measurements (Haghighi et al., 2018) are adopted to establish CSM, which captures the interconnectedness between SM and EF. Nonetheless, the EF-SM method has limitations due to sparse H and LE site observations (Feldman et al., 2019; Fu et al., 2022a), which poses challenges in adequately capturing comprehensive regional or continental-scale CSM variations that arise from different vegetation and soil conditions (Van Looy et al., 2017).





In recent years, the feasibility of conducting large-scale analysis has been enhanced by the growing accessibility of multi-source satellite-based datasets (Liu et al., 2012). Akbar et al. (2018) conducted an assessment of dry-down features across the contiguous United States based on satellite surface SM data obtained from the National Aeronautics and Space Administration Soil Moisture Active Passive mission (SMAP). Specifically, SM exhibits a temporary increase immediately after the rain event, followed by a gradual decline until the occurrence of the next rainfall event, known as dry-down (Akbar

et al., 2018; Feldman et al., 2019). Feldman et al. (2019) used a piecewise linear model to estimate CSM throughout Africa based on satellite-derived data on surface SM and diurnal temperature amplitude. The advancement of global remote sensing products technology has facilitated the generation of reliable GPP products (Yuan et al., 2014; Li and Xiao, 2019; Zhang et al., 2019; Bi et al., 2022; He et al., 2022; Li, F. et al., 2023) and ET products (Yao et al., 2013; Yao et al., 2014; Zhang et al., 2019; Cheng et al., 2021; He et al., 2022; Li, C. et al., 2022; Li, F. et al., 2023). Denissen et al. (2020) proposed a new

tipping-point metric using ET and SM to straightforwardly determine CSM at the continental scale. Coupled with the ET process via plant leaf stomata, GPP at the ecosystem scale is strongly connected to ET (Gentine et al., 2019; Liu, Y. et al., 2020) and also associated with the CSM value as identified by Champagne et al. (2012). As plants adjust their stomatal resistance in response to changes in water stress and energy requirements in the form of VPD (Grossiord et al., 2020; Li, F. et al., 2023). Fu et al. (2022a) first demonstrated that the covariance between GPP and VPD indirectly quantifies CSM. The

point at which covariance between GPP and VPD transitions from positive to negative during a period of soil drying is denoted as CSM. However, it is not clear whether GPP occurs in water and energy limit shifts at the same CSM as ET. In addition, there are significant differences among satellite-based ET and GPP datasets and the uncertainties of CSM vary with different methods. A source of considerable uncertainty in considering only a single data source and estimation approach exists at a large spatial scale.

Chinese land surface frequently experiences water and energy limit shifts (Xiao, 2014; Zhu et al., 2023). As such, this study uses two satellite-based innovative metrics to assess the characteristics of the CSM over China: the covariance between GPP and VPD (VPD-GPP-SM method); the difference between the correlation between SM and ET and the correlation between VPD and ET (correlation-difference method). The goal of this study is to: (1) assess the CSM performance using site observations through the VPD-GPP-SM and correlation-difference methods compared to that through the EF-SM method; (2)

examine CSM variations using satellite-based products across land cover types, soil textures, and water resource subregions; and (3) investigate dominant factors from climate and ecosystem variables that influence CSM.

## 2 Material and methods

### 2.1 Data

The flux site-based datasets were used for the assessment of satellite-based ET and GPP in Section 3.1 and CSM derived

from the VPD-GPP-SM and correlation-difference methods in Section 3.2 at the site scale. The layer-wise SM and satellite-based ET and GPP were used for the large-scale detection of CSM. Soil texture, land cover type, and water resource





regionalization were all used to examine CSM variations in Section 3.3. The SM, ET, GPP, and meteorological data were all used to investigate the dominant factors that influence CSM in Section 3.4. In order to facilitate comparability, all energy, vegetation, and water variables were resampled or combined to 0.1°-8 days resolution. The time span, limited by the temporal availability of several data sources, covered the period 2001–2018. The detailed information on flux sites is shown in Table 1. Table 2 contains a list of all spatial data sets used in this study.

### 2.1.1 Water resource regionalization

In order to conduct a thorough examination of regional CSM, the research area was partitioned into 15 subregions pertaining to water resources (Fig. 1), using the classification system established by the China Institute of Geo-Environment Monitoring. The following are the names of various basins and regions: Zhungaer Basin (ZGE), Pearl River Basin (PR), Yangtze River Basin (YTR), Southwestern River Basin (SWR), Tarim Basin (TR), Songhua River Basin (SR), Changthang Region (CT), Inner Mongolian Plateau Region (NM), Liaohe River Basin (LR), Yellow River Basin (YR), Huaihe River Basin (HR), Hexi Corridor Region (HC), Haihe River Basin (HAR), Southeastern River Basin (SER), and Qaidam Basin (QB). The regionalization of water resources is based on the principles of groundwater systems and water cycles. The suggested concepts for subregions are focused on the inherent features of groundwater resources within distinct natural units.

### 2.1.2 Site observations

Figure 1 illustrates the locations of the flux sites. Eddy covariance observed in-situ ET and GPP were used to evaluate the performance of gridded satellite-based ET and GPP. Flux site-observed temperature (Ta), incoming shortwave radiation (Rs), VPD, EF, SM, ET, and GPP were used to evaluate the performance of CSM derived from the VPD-GPP-SM and correlation-difference methods at the site scale. Three flux networks were used including the National Tibetan Plateau Data Center (TPDC), ChinaFlux, and Fluxnet. TPDC at the http://data.tpdc.ac.cn/ website offers data access to a total of eight flux sites. These sites consist of four locations inside the HC (Huazhaizi, Dashalong, Luodi, and Arou) and four cropland sites within the HAR (Guantao, Huailai, Miyun, and Daxing). The data is recorded at half-hour intervals. ChinaFlux offers site-observed data from Damshung, Xilingela, Xishuangbanna, Dinghushan, Qianyanzhou, Changbaishan, Yucheng, Haibei1, and Haibei2 flux sites. These data can be obtained from http://www.chinaflux.org/. Fluxnet includes four grassland site, CN-Sw2, CN-Du2, CN-Du3, and CN-Cng with daily records, which is available at https://fluxnet.org/data/download-data/. Given the fact that the TPDC did not have GPP data, the REddyProc was used to compute GPP using the water and carbon fluxes at Huazhaizi, Dashalong, Luodi, Arou, Guantao, Huailai, Miyun, and Daxing sites

**Figure 1: (a) Locations of the flux sites, land cover types (2020), and fifteen water resource subregions of China. Distributions of (b) clay, (c) silt, and (d) sand content (1995). ZGE: Zhungaer Basin, PR: Pearl River Basin, YTR: Yangtze River Basin, SWR: Southwestern River Basin, TR: Tarim Basin, SR: Songhua River Basin, CT: Changthang Region, NM: Inner Mongolian Plateau**





**Region, LR: Liaohe River Basin, YR: Yellow River Basin, HR: Huaihe River Basin, HC: Hexi Corridor Region, HAR: Haihe River Basin, SER: Southeastern River Basin, QB: Qaidam Basin.**

**Table 1: Flux site information used in this study.**

| Site | Land cover types | Latitude | Longitude | Time span | Source |
|---|---|---|---|---|---|
| CN-Sw2 | | 41.79 | 111.89 | 2011 | Fluxnet |
| CN-Du2 | | 42.04 | 116.28 | 2006–2008 | Fluxnet |
| CN-Du3 | | 42.05 | 116.28 | 2009–2010 | Fluxnet |
| CN-Cng | | 44.59 | 123.51 | 2007–2010 | Fluxnet |
| Damshung | Grassland | 30.49 | 91.06 | 2004–2010 | Chinaflux |
| Xilingela | | 43.53 | 116.67 | 2004–2010 | Chinaflux |
| Haibei1 | | 37.37 | 101.18 | 2003–2010 | Chinaflux |
| Dashalong | | 38.84 | 98.94 | 2013–2015 | TPDC |
| Arou | | 38.04 | 100.46 | 2013–2015 | TPDC |
| Daxing | | 39.62 | 116.43 | 2008–2010 | TPDC |
| Miyun | | 40.63 | 117.32 | 2008–2009 | TPDC |
| Huailai | Cropland | 40.35 | 115.79 | 2014–2018 | TPDC |
| Guantao | | 36.52 | 115.13 | 2008–2009 | TPDC |
| Yucheng | | 36.82 | 116.57 | 2003–2010 | Chinaflux |
| Xishuangbanna | Evergreen broadleaf forests | 21.92 | 101.26 | 2003–2010 | Chinaflux |
| Dinghushan | | 23.16 | 112.53 | 2003–2010 | Chinaflux |
| Qianyanzhou | Evergreen needleleaf forests | 26.74 | 115.05 | 2003–2010 | Chinaflux |
| Changbaishan | Mixed forests | 42.40 | 128.09 | 2003–2010 | Chinaflux |
| Haibei2 | Wetland | 37.66 | 101.33 | 2004–2009 | Chinaflux |
| Huazhaizi | barren | 38.76 | 100.32 | 2013–2015 | TPDC |
| Luodi | | 41.99 | 101.13 | 2014–2015 | TPDC |

TPDC: National Tibetan Plateau Data center

### 2.1.3 Soil moisture

Given the recent availability of state-of-the-art gridded SM in China released by Li, Q. et al. (2022), CSM can now be investigated in the context of the SM state. Soil Moisture of China by in situ data, version 1.0 (SMCI1.0) is available in

TPDC. The data comprises gridded datasets reaching 100 cm soil depth with 10 cm intervals at 1 km-daily resolution during 2000–2020. The robust random forest machine learning technique was used to train the predictors of ERA5-Land time series, leaf area index (LAI), land cover type, topography, and soil attributes using in situ observations from 1789 stations throughout China. Based on the findings of Li, Q. et al. (2022), SMCI1.0 product demonstrates notable benefits over both the ERA5-Land and SMAP-L4 datasets, especially in terms of a superior quality level compared to the SoMo.ml dataset at

soil depths of 10, 20, 80, and 100 cm. Thus, this study utilized SM at soil depths of 10, 20, 80, and 100 cm.



### 2.1.4 Satellite-based ET and GPP

Advances in remote sensing have substantially fostered the development of global ET and GPP products for CSM simulation. Four ET products and four GPP products were employed (Table 2), including Penman-Monteith-Leuning (PML) ET and GPP, Global LAnd Surface Satellite (GLASS) ET and GPP, Collocation-Analyzed Multi-source Ensembled Land

Evapotranspiration (CAMELE) ET, Surface Energy Balance Algorithm for Land evapotranspiration in China (SEBAL) ET, Two-Leaf light use efficiency (LUE) model based (TL) GPP, and Global, Orbiting Carbon Observatory-2 (OCO-2) SIF-based (GOSIF) GPP.

PML provides ET and GPP for the Chinese region with a spatiotemporal resolution of 500 m and 1 day during February 2000–December 2020 on TPDC website. It integrates the stomatal conductance theory to relate the GPP process using the

Penman-Monteith-Leuning model (Zhang et al., 2019; He et al., 2022) and applies daily meteorological data, land surface temperature from ERA5, enhanced Whittaker-filtered MODIS LAI, albedo, and emissivity. The interdependency and mutual restrictions between GPP and ET considerably increase the accuracy of ET simulation.

GLASS ET and GPP are provided at 0.05° resolution and every 8 days by http://glass.umd.edu/. GLASS ET integrates the MOD16, a revised remote sensing-based Penman-Monteith, the Priestley-Taylor Jet Propulsion Laboratory, a modified

satellite-based Priestley-Taylor, and the Semi-Empirical Algorithm of the University of Maryland using Bayesian Model Averaging (BMA) approach (Yao et al., 2013, Yao et al., 2014). GLASS GPP algorithm incorporates the effects of atmospheric carbon dioxide content, radiation components, and VPD based on the Eddy Covariance-LUE model introduced by Yuan et al., (2007). It is founded on two underlying assumptions: the fraction of absorbed photosynthetically active radiation (fPAR) has a linear relationship with the normalized difference vegetation index (NDVI); constant LUE is

governed by either air temperature or soil moisture, depending on which component imposes the greatest limitation.

CAMELE ET provides a long-term (1981–2020) collocation-analyzed multi-source ensembled land ET, employing ERA5, FLUXCOM, PML, GLDAS, and GLEAM (Li, C. et al., 2022), at 0.1°-8 days and 0.25°-daily resolutions on Zenodo: https://zenodo.org/record/6283239.

SEBAL ET focuses on 1 km-daily resolution during 2001–2018 and is publicly accessible from the Zenodo repository at

https://doi.org/10.5281/zenodo.4243988 and https://doi.org/10.5281/zenodo.4896147. This product integrates GMAO's meteorological data and NASA's MOD43A1 daily surface albedo, MOD11A1 daily surface temperature, and MOD13 vegetation index (Cheng et al., 2021).

TL GPP (https://doi.org/10.5061/dryad.dfn2z352k) offers comprehensive worldwide assessments of GPP, shaded GPP, and sunlit GPP at a spatiotemporal resolution of 0.05°-8 days, covering the period from 1992 to 2020. A two-leaf LUE model is

used with recent data inputs such as the GLOBMAP LAI, CRUJRA meteorological data, and ESA-CCI land cover information (Bi et al., 2022).

GOSIF GPP spans from 2000 to 2020 with 0.05°-8 days resolution and is obtained from https://globalecology.unh.edu/. A total of eight SIF-GPP relationships including both universal and biome-specific formulations were used to estimate GPP





from SIF on a per-pixel basis. These relationships were examined with and without intercept terms to account for the
uncertainty in converting SIF from the OCO-2 SIF to the GPP estimates (Li and Xiao, 2019).

### 2.1.5 Meteorological data, land cover types, and soil texture

Yang et al. (2010) and He et al. (2020) put forth a comprehensive dataset for Chinese regional surface meteorological
forcing. This dataset encompasses air temperature, air pressure, specific humidity, wind speed, downward shortwave
radiation, downward longwave radiation, and precipitation. It is presented in the NetCDF format with a spatiotemporal
resolution of 0.1°-3 hours during 1979–2018. The primary data sources include international Princeton reanalysis data,
GLDAS data, GEWEX-SRB radiation data, TRMM precipitation data, and meteorological from the China Meteorological
Administration. Data quality control techniques include the elimination of physically implausible values and statistical
interpolation using ANU-Spline. In addition, this dataset demonstrates precision levels that lie between those of site-based
observation and satellite-based estimation, therefore exceeding the accuracy of current international reanalysis datasets. In
this study, specific humidity and air temperature were used to compute VPD, while precipitation and downward shortwave
radiation were employed in the examination of water and energy limitations.

Land cover types and soil textures were contributed by Data Center for Resources and Environmental Sciences, Chinese
Academy of Sciences (RESDC) (http://www.resdc.cn). Land cover types (2020) were created by human visual interpretation
relying on Landsat satellite remote sensing images. It utilized a categorization scheme including cropland, forests, grassland,
water bodies, urban, and barren. Soil textures were compiled from the 1:1,000,000 soil type map and the second national soil
survey. It is expressed as sand, silt, and clay content within each grid cell to accurately depict distinct texture types.

**Table 2: Spatial gridded data sets used in this study.**

| Variable | Dataset | Spatial resolution | Temporal resolution | Unit | Time span | Reference |
|---|---|---|---|---|---|---|
| Soil moisture | SMCI1.0 | 0.1° | day | 0.001$m^3$/$m^3$ | 2000–2020 | Li, Q. et al. (2022) |
| | GLASS | 0.05° | 8-day | W/$m^2$ | 2000–2018 | Yao et al. (2013, 2014) |
| Evapotranspiration | PML | 500 m | day | 0.01mm | 2000–2020 | Zhang et al. (2019) and He et al. (2022) |
| | CAMELE | 0.1° | 8-day | kg/ $m^2$ /s | 2001–2019 | Li, C. et al. (2022) |
| | SEBAL | 1 km | day | mm | 2001–2018 | Cheng et al. (2021) |
| | GLASS | 0.05° | 8-day | | 1982–2018 | Yuan et al. (2014) |
| Gross primary production | PML | 500 m | day | gC/$m^2$ | 2000–2020 | Zhang et al. (2019) and He et al. (2022) |
| | GOSIF | 0.05° | 8-day | | 2000–2021 | Li and Xiao (2019) |
| | TL | 0.05° | 8-day | | 1992–2020 | Bi et al. (2022) |
| Specific humidity Air temperature | - | 0.1° | 3-hour | kg kg$^{-1}$ K | 1979–2018 | Yang et al. (2010) and He et al. (2020) |





| | | | | | | |
|---|---|---|---|---|---|---|
| Downward shortwave radiation | | | | W m$^{-2}$ | | |
| Precipitation | | | | mm hr$^{-1}$ | | |
| Land cover | - | 1 km | - | - | 2020 | http://www.resdc.cn |
| Soil texture | - | 1 km | - | - | 1995 | http://www.resdc.cn |

## 2.2 Determination of CSM

CSM derived by the EF-SM method was used to assess the CSM from ET and GPP on the site scale. To assess CSM during

water and energy limit shifts, there must be both positive and negative metrics from the VPD-GPP-SM and correlation-difference methods. The data will be taken into account just when the temperature surpasses 10° (Denissen et al., 2020) to avoid the influence of ice and snow, and the covariance between VPD and GPP must exhibit a minimum of seven covariance values within 9-day moving windows, with a minimum of 15 data (Fu et al., 2022a). Hence, we concentrated on the warm season during June–September including 16 data each year with 8 values over 9 moving windows. CSM using satellite-based

ET and GPP were conducted in each grid cell.

### 2.2.1 EF-SM method

A linear-plus-plateau model was applied to characterize the relationship between EF and SM precisely measured by eddy covariance flux towers (Seneviratne et al., 2010; Schwingshackl et al., 2017):

$$EF = \begin{cases} EF_{max} + S(SM - CMS), & \text{if } SM < CSM \\ EF_{max}, & \text{if } SM \geq CSM \end{cases}, \tag{1}$$

where EF_max represents the maximum EF in the energy-limited stage, and S is the gradient in the water-limited stage. Site-specific estimated CSM was simultaneously estimated by the Monte Carlo (MC) method. For a set of optimal parameters, the Nash-Sutcliffe model efficiency coefficient (NSE) was used as the fitness function (Nash and Sutcliffe,1970).

### 2.2.2 VPD-GPP-SM method

The VPD-GPP-SM approach presents a novel method for assessing ecosystem water stress in direct correlation with GPP as

illustrated by Fu et al. (2022a). It serves to quantify CSM over large areas. The positive covariances between VPD and GPP indicate that energy limits GPP. The presence of negative covariances indicates that water limitation has a larger impact on GPP. VPD is determined by the disparity between the saturation vapor pressure (es) and the actual vapor pressure (ea). Bolton (1980) posits that the calculation of ea involves the use of specific humidity (SH) and surface pressure (Pr):

$$e_a = \frac{SH \times Pr}{SH \times 0.378 + 0.622}, \tag{2}$$




### 2.2.3 Correlation-difference method

Another novel correlation-difference metric, Corr(ET, VPD)-Corr(ET, SM), proposed by Denissen et al. (2020), evaluates water versus energy-limited conditions using linearly detrended VPD, ET, and SM anomalies. Matlab's corr Tool calculates this metric, which uses Kendall's rank correlation (Corr) rather than assuming linear correlations between variables (van Doorn et al., 2018). If Corr(ET, VPD)-Corr(ET, SM) > 0, then the grid cell is energy-limited and vegetation anomalies (i.e., ET) correlate more strongly with energy anomalies (i.e., VPD) than with water anomalies (i.e., SM). Corr(ET, VPD)-Corr(ET, SM) < 0, in contrast, is water-limited. When Corr(ET, VPD)-Corr(ET, SM) ≈ 0, SM is labeled as CSM, indicating that water and energy limit regimes are transitioning. Next, Ta and Rs were used for site-based CSM to reveal the performance of VPD, Ta, and Rs in this method.

### 2.3 Evaluation criteria

Correlation coefficient was calculated to evaluate the performance of satellite-based ET from CAMELE, GLASS, PML, and SEBAL and GPP from GOSIF, GLASS, PML, and TL, compared to the eddy covariance observed in-situ ET and GPP. A point-to-pixel evaluation was carried out to evaluate the over or underestimation of ET and GPP for each land cover type from all 21 flux sites. We summed 8-day ET and GPP changes in grassland (GL), evergreen broadleaf forests (EBF), evergreen needleleaf forests (ENF), mixed forests (MF), cropland (CL), wetland (WL), and barren land.

NSE was utilized to evaluate the relationship between EF and SM, and values above 0.5 were considered satisfactory (Herman et al., 2018). Only 9 sites with NSE values above 0.5, including Xilingela in 2004, Damshung in 2004, CN-Sw2 in 2011, CN-Du2 in 2007, CN-Du3 in 2009, CN-Cng in 2010, Miyun in 2009, Huailai in 2015, and Qianyanzhou in 2010, were applied for CSM detection.

### 2.4 Ridge regression

Ridge regression has been widely acknowledged as a viable approach for mitigating collinearity issues among independent variables (Liu et al., 2021; Zhu et al., 2022). This technique has found extensive use in quantifying the interannual impacts of climate and plant growth variations on water and energy dynamics. Here, the dominant factor of precipitation, Rs, ET, GPP, and SM on Corr(ET, VPD)-Corr(ET, SM) in June–September over the period 2001–2018 was identified by the largest contribution obtained by multiplying the slope of the normalized independent component with the ridge regression coefficient.





## 3 Results

### 3.1 Assessment of satellite-based ET and GPP

Figure 2a and b show good agreement between ET and GPP derived from the satellite-based products and site observations in most of the land cover types. Over seven land cover types, correlation coefficients obtained from CAMELE, GLASS,

PML, and SEBAL ET were 0.74, 0.65, 0.78, and 0.59, respectively; correlation coefficients obtained from GLASS, TL, GOSIF, and PML GPP were 0.75, 0.71, 0.77, and 0.74, respectively. For specific land cover types, the highest correlation coefficient occurred between GLASS ET and eddy covariance observations in MF (0.96), while the lowest value was between SEBAL and site observations in barren (0.47). The highest correlation coefficient was found between TL GPP and site measurements in MF (0.97), while the lowest value was between GLASS GPP and site-based data in barren land (0.32).

In general, no single product consistently outperformed the others over all types.

Figure 2c and d show the comparison between daily site observations and satellite-based ET and GPP across land cover types. ET from the products had the highest value in ENF and was the lowest in barren land, while GPP estimate peaked in EBF and was the lowest in WL. The site-observed ET was 1.91 mm in ENF and 0.78 mm in barren land. The site-observed GPP in EBF and WL were 5.26 and 1.34 gC m$^{-2}$ day$^{-1}$, respectively. In these land cover types, ET and GPP derived from

satellite-based products were also substantially different and varied quite a bit between different products. Especially in EBF, ET derived from GLASS (3.37 mm) and CAMELE (3.05 mm) and GPP from GOSIF (7.55 gC m$^{-2}$ day$^{-1}$) and TL (7.59 gC m$^{-2}$ day$^{-1}$) were higher than site observations of 1.74 mm and 5.26 gC m$^{-2}$ day$^{-1}$, respectively. If satellite-based ET and GPP were between ±10% of site-observed values, they were termed as satisfactory; otherwise, they were either over or underestimated. CAMELE, GLASS, PML, and SEBAL ET and GLASS, TL, GOSIF, and PML GPP met the satisfied values

in 1, 1, 3, 1, 2, 1, 3, and 2 land cover types, respectively. PML ET provided the most satisfactory estimates in EBF, CL, and barren land with an average bias of 1.05%, 1.13%, and 1.34%, respectively; GOSIF GPP provided the most satisfactory estimates in GL, ENF, and MF with an average bias of 4.31%, 9.14%, and 4.29%, respectively. Although discrepancies existed among multi-source remotely sensed products across flux sites, they offered an opportunity to quantify the characteristics of large-scale CSM and examine the uncertainty from a single source data.







**Figure 2:** Correlation coefficients between eddy covariance-based observations and satellite-based (a) ET and (b) GPP products across land cover types, GL: grassland, EBF: evergreen broadleaf forests, ENF: evergreen needleleaf forests, MF: mixed forests, CL: cropland, WL: wetland. Comparison of the daily (c) ET from CAMELE, GLASS, PML, and SEBAL and (d) GPP from GLASS, TL, GOSIF, and PML with the measurements with their associated 95% confidence intervals across land cover types.

## 3.2 Assessment of CSM derived from the VPD-GPP-SM and correlation-difference methods

The variations of EF and SM were depicted in Fig. 3a, c, e, g, i, k, m, o, q for nine sites. The fitted lines represented the controlling mechanisms in the various evaporative regimes. Overall, EF increased with increasing SM during water-limited regimes. Specifically, grassland in CN-Du2 showed a greater slope at low SM values with an NSE of 0.82 than cropland in





Miyun with an NSE of 0.51. We also found that the CSM of grassland in different regions varied greatly. Grassland in

Xilingela had the lowest CSM of 0.07 $m^3/m^3$. CSM in the grassland Damshung in Southwest China was 0.15 $m^3/m^3$ with SM ranging from 0.14 to 0.26 $m^3/m^3$. CSM in CN-Cng in Northeast China was 0.39 $m^3/m^3$ with high SM ranging from 0.30 to 0.70 $m^3/m^3$. The ranges of SM across land cover types determined the CSM value.

Moreover, we evaluated CSM from the VPD-GPP-SM and correlation-difference methods against the reference CSM from the EF-SM method. Corr(ET, VPD)-Corr(ET, SM), Corr(ET, Ta)-Corr(ET, SM), and Corr(ET, Rs)-Corr(ET, SM) using the

correlation-difference method and Cov(VPD, GPP) using covariance between VPD and GPP were in line with that shown in Fig. 3b, d, f, h, j, l, n, p, r for nine sites. Vertical lines of different colors represented the CSM derived from one of the four methods mentioned above, which showed the best agreement with the CSM derived from the EF-SM method. We found that the tipping point of Corr(ET, VPD)-Corr(ET, SM) had good consistent with that of EF-SM, especially in Xilingela and Huailai sites, which changed from negative (water-limit) to positive (energy-limit). Through another method, CSM detected

by Cov(VPD, GPP) was consistent with that from EF-SM over four sites, especially in CN-Du2 and CN-Du3. Table 3 shows CSM values detected by different methods at nine sites. For CSM from the correlation-difference method using VPD, Ta, and Rs, it was found that CSM using Ta, Corr(ET, Ta)-Corr(ET, SM), had only one optimal CSM value that agreed best with the EF-SM derived CSM at Miyun, indicating the lowest detection ability compared to other methods. Among CSM using VPD and Rs, it was found that CSM using Rs detected no CSM value at three sites (Xilingela, Miyun, and Huailai) while

using VPD detected no CSM value at only two sites (CN-Sw2 and Miyun) indicating its better performance than the former. Thus, Corr(ET, VPD)-Corr(ET, SM) was the best among the three methods. In addition, Cov(VPD, GPP) had the four optimal CSMs that agreed best with the EF-SM derived CSM at Damshung, CN-Du2, CN-Du3, and CN-Cng. Therefore, Corr(ET, VPD)-Corr(ET, SM) and Cov(VPD, GPP) had the potential to obtain large-scale CSM.





**Figure 3: Quantifying the critical soil moisture (CSM) using the evaporative fraction and soil moisture (EF-SM) method at (a) Xilingela in 2004, (c) Damshung in 2004, (e) CN-Sw2 in 2011, (g) CN-Du2 in 2007, (i) CN-Du3 in 2009, (k) CN-Cng in 2010, (m) Miyun in 2009, (o) Huailai in 2015, (q) Qianyanzhou in 2010. CSM using covariance (referred to as Cov(VPD, GPP)) between vapor pressure deficit (VPD) and gross primary production (GPP), and correlation-difference metric with Kendall's rank correlation (KC) between detrended anomaly evapotranspiration (ET) and SM (referred to as Corr(ET, SM)) and KCs between detrended anomaly ET and VPD (referred to as Corr(ET, VPD)), air temperature (referred to as Corr(ET, Ta)), and incoming**





**shortwave radiation (referred to as Corr(ET, Rs)) at (b) Xilingela in 2004, (d) Damshung in 2004, (f) CN-Sw2 in 2011, (h) CN-Du2 in 2007, (j) CN-Du3 in 2009, (l) CN-Cng in 2010, (n) Miyun in 2009, (p) Huailai in 2015, (r) Qianyanzhou in 2010.**

**Table 3: CSM obtained from five methods as described in Fig. 3.**

| Site | CSM from EF-SM | Cov(VPD, GPP) | Corr(ET, VPD)- Corr(ET, SM) | Corr(ET, Ta)- Corr(ET, SM) | Corr(ET, Rs)- Corr(ET, SM) |
|---|---|---|---|---|---|
| Xilingela | 0.078 | - | **0.079\*** | - | - |
| Damshung | 0.153 | **0.172\*** | 0.246 | - | 0.243 |
| CN-Sw2 | 0.135 | 0.116 | - | - | **0.131\*** |
| CN-Du2 | 0.107 | **0.108\*** | 0.147 | 0.101 | 0.205 |
| CN-Du3 | 0.230 | **0.219\*** | 0.190 | 0.163 | 0.273 |
| CN-Cng | 0.391 | **0.380\*** | 0.420 | 0.404 | 0.561 |
| Miyun | 0.271 | - | - | **0.319\*** | - |
| Huailai | 0.194 | 0.223 | **0.190\*** | 0.185 | - |
| Qianyanzhou | 0.140 | - | 0.115 | 0.115 | **0.122\*** |

*: value in bold agrees best with the EF-SM derived CSM

### 3.3 Spatial pattern of CSM derived from the VPD-GPP-SM and correlation-difference methods

Figure 4 shows the spatial distribution of CSM obtained from covariance between VPD and GOSIF, GLASS, PML, and TL GPP, and correlation-difference metric with Kendall's rank correlation (KC) between detrended anomaly CAMELE, GLASS, PML, SEBAL ET and 10, 20, 80, and 100 cm soil depths SM, and KC between detrended anomaly ET and VPD. Geographically, they spanned large swaths of land through water-scarce desert regions and lush, rainy forests. The intercomparison provided helpful insights to examine the consistency and discrepancy between multi-source ET and GPP products in depicting the spatial distribution of CSM. Overall, the spatial patterns of CSM obtained through the four ET products were consistent with those from the four GPP products, showing a decreasing variation from South to North China. Specifically, GLASS GPP displayed no CSM value in southeastern YTR. Additionally, a layer-wise CSM analysis was conducted to highlight variations in SM properties for different soil layers. Spatial patterns of CSM changed with increasing layer depth. For 20 cm soil depth, CSM in TR decreased. For 100 cm soil depth, CSM in some north parts of HAR decreased from 0.3–0.4 to 0.2–0.3. Hence, it was evident that there was a great degree of variation in the CSM behavior across layers, especially in TR and HAR. Table 4 shows the comparison of site CSM from EF-SM and gridded CSM using satellite-based ET and GPP and 10 cm depth SM. It was found that gridded CSMs in CN-Du3, CN-Cng, Miyun, and Huailai were generally consistent with site-based values. Gridded data had spatial continuity and site observations showed significant differences in CSM even between adjacent sites (e.g., CN-Du3 of 0.11 m³/m³ and CN-Du3 of 0.23 m³/m³), resulting in inconsistent CSM between satellite and site-based value.

Specifically for water resources subregions (Fig. 5), HC showed a decrease in SM as soil depth increased. In explaining moisture thresholds between water and energy-limited regimes, CSM in semi-humid HAR and HR was about 0.27 m³/m³ and 0.29 m³/m³, respectively, and increased to approximately 0.40 m³/m³ in humid South China (SER and PR). We also





found that there was little difference between the four layers of CSM in South China regions like PR and SER mainly composed of forests (both around 0.40 m³/m³). However, in addition to the decrease in CSM in the HC from 0.17 m³/m³ at 10 cm soil depth to 0.14 m³/m³ at 100 cm, there had been a widespread decrease in CSM in the basins north of the YTR mainly composed of grass and crop. Root depths might lead to differences in soil vertical CSM. Furthermore, large-scale CSM varied across vegetation types and soil textures (Fig. 6). With shorter root systems and less vegetation (i.e., barren),

areas with low CSM were water-limited. Forested regions displayed a relatively high CSM (e.g., 0.37 m³/m³ using GOSIF GPP and 10 cm depth SM). As for the soil types, sand covering the large area was further part into content of less than 60%, 60–70%, 70–80%, 80–90%, and higher than 90%. Soils with a majority of clay had a wetter CSM than others (e.g., 0.39 m³/m³ using GOSIF GPP and 10 cm depth SM) and was to be expected given that clay had a larger negative matric potential compared to coarse soil textures dominated by sand and silt. Higher CSM indicated wetter soil and hence limitation by

energy. In summary, fine soils and luxuriant vegetation had wetter CSM.





CSM using GOSIF GPP     CSM using GLASS GPP     CSM using PML GPP     CSM using TL GPP

(a)     (b)     (c)     (d)

CSM using CAMELE ET     CSM using GLASS ET     CSM using PML ET     CSM using SEBAL ET

(e)     (f)     (g)     (h)

10 cm

(i)     (j)     (k)     (l)

20 cm

(m)     (n)     (o)     (p)

80 cm

(q)     (r)     (s)     (t)

100 cm

CSM

0.04   0.1   0.2   0.3   0.4

**Figure 4: Quantifying the spatial pattern of critical soil moisture (CSM) at 10 cm depth using covariance between vapor pressure deficit (VPD) and gross primary production (GPP) from (a) GOSIF, (b) GLASS, (c) PML, (d) TL. CSM using correlation-difference metric with Kendall's rank correlation (KC) between detrended anomaly soil moisture (SM) at (e–h) 10 cm, (i–l) 20 cm,**





**(m–p) 80 cm, and (q–t) 100 cm depths depth and evapotranspiration (ET) from CAMELE, GLASS, PML, and SEBAL and KC between detrended anomaly VPD and those ET products.**

**Table 4: Site CSM from EF-SM and gridded CSM using satellite-based ET and GPP and 10 cm depth SM.**

| Site | CSM from EF-SM | CSM using GOSIF GPP | CSM using GLASS GPP | CSM using PML GPP | CSM using TL GPP | CSM using CSMELE ET | CSM using GLASS ET | CSM using PML ET | CSM using SEBAL ET |
|---|---|---|---|---|---|---|---|---|---|
| Xilingela | 0.078 | 0.263 | 0.261 | 0.263 | 0.254 | 0.260 | 0.260 | 0.256 | 0.265 |
| Damshung | 0.153 | 0.385 | 0.382 | 0.383 | 0.381 | 0.381 | 0.380 | 0.378 | 0.377 |
| CN-Sw2 | 0.135 | 0.255 | 0.269 | 0.251 | 0.236 | 0.256 | 0.270 | 0.258 | 0.254 |
| CN-Du2 | 0.107 | 0.276 | 0.289 | 0.274 | 0.264 | 0.281 | 0.286 | 0.288 | 0.288 |
| CN-Du3 | 0.230 | 0.276 | 0.289 | 0.274 | **0.264*** | 0.281 | 0.286 | 0.288 | 0.288 |
| CN-Cng | 0.391 | 0.336 | **0.357*** | 0.339 | 0.329 | 0.355 | 0.322 | 0.353 | 0.342 |
| Miyun | 0.271 | 0.323 | 0.336 | 0.327 | 0.323 | 0.327 | **0.320*** | 0.346 | 0.331 |
| Huailai | 0.194 | 0.270 | 0.265 | 0.251 | **0.238*** | 0.257 | 0.260 | 0.268 | 0.263 |
| Qianyanzhou | 0.140 | 0.411 | - | 0.409 | 0.384 | 0.391 | 0.422 | 0.403 | 0.420 |

*: value in bold agrees best with the EF-SM derived CSM











**Figure 5: Soil moisture (SM) at (a) 10 cm, (b) 20 cm, (c) 80 cm, and (d) 100 cm soil depths across China, and critical soil moisture (CSM) derived from CAMELE, GLASS, PML, and SEBAL ET and SM at corresponding soil depths and CSM obtained from TL, GLASS, GOSIF, and PML GPP and 10 cm soil depth SM. ZGE: Zhungaer Basin, PR: Pearl River Basin, YTR: Yangtze River Basin, SWR: Southwestern River Basin, TR: Tarim Basin, SR: Songhua River Basin, CT: Changthang Region, NM: Inner Mongolian Plateau Region, LR: Liaohe River Basin, YR: Yellow River Basin, HR: Huaihe River Basin, HC: Hexi Corridor Region, HAR: Haihe River Basin, SER: Southeastern River Basin, QB: Qaidam Basin.**

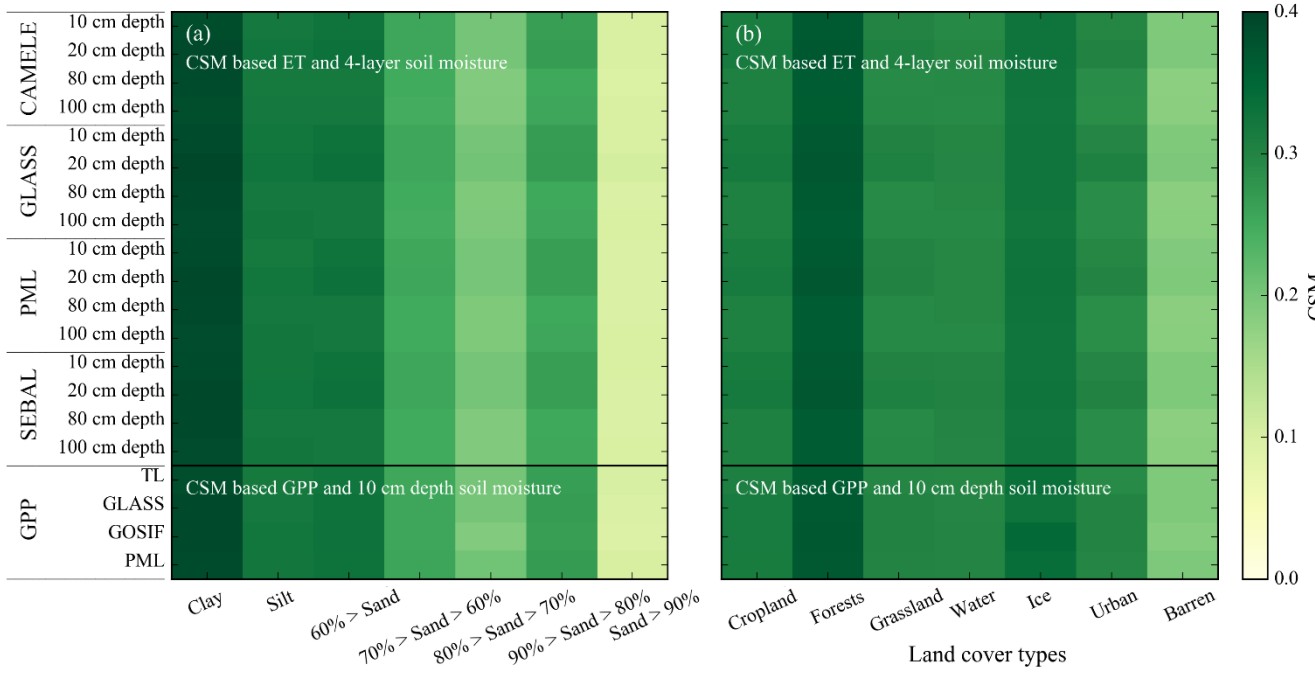

**Figure 6: Critical soil moisture (CSM) derived from ET (CAMELE, GLASS, PML, and SEBAL) and soil moisture (SM) (10, 20, 80, and 100 cm soil depths) and CSM obtained from GPP (TL, GLASS, GOSIF, and PML) and 10 cm soil depth SM for (a) soils with a majority of clay, silt, and sand with content less than 60, between 60% and 70%, between 70% and 80%, between 80% and 90%, and higher than 90%, (b) cropland, forests, grassland, water, ice, urban, and barren.**

### 3.4 Attribution of water and energy limit shifts

We assessed the mean and the slope of Corr(ET, VPD)-Corr(ET, SM) using ET, VPD, and 10 cm soil depth SM in June–September over the period 2001–2018. PML ET was used for analysis given the fact that it had the best performance (Section 3.1). As shown in Fig. 7a, the water-limited regimes were most common in dry and semi-arid areas. Western and northern regions were generally water-limited, especially in CT. Southern regions including YTR and SER were energy-limited. We also found that areas currently lacking water in western and northern China encountered decreased energy limitation, thereby intensifying water scarcity, especially in parts of HAR, YR, HC, QB, ZGE, TR, and CT (Fig. 7b). Decrease Corr(ET, VPD)-Corr(ET, SM) in these areas were more widespread than increase. In contrast, increases in Corr(ET, VPD)-Corr(ET, SM) were found across YTR, HR, and SR. In addition, the southern regions experienced decreased energy limitation.





Attributing Corr(ET, VPD)-Corr(ET, SM) variations to hydrological, meteorological, and ecological predictors, we identified five factors (P, Rs, ET, GPP, and SM) using ridge regression. GOSIF GPP was used for analysis given the fact that it had the best performance (Section 3.1). As shown in Fig. 8, ET was dominant in water-limited western areas, such as TR with an area per region of 44% and CT with an area per region of 49%. Decreased energy limitation in these regions was

dominated by increasing ET. GPP was the most important in terms of the northeastern regions, such as SR (43%), LR (48%), YR (47%), HAR (50%), and HR (52%). Increased energy limitation in these regions was associated with increasing GPP, while Rs was dominant in the transition between different regimes. In energy-limited regimes, the primary mechanism that the surface responsiveness to forcing spatial pattern was that SM dominated Corr(ET, VPD)-Corr(ET, SM) for northwest ZGE (30%), northern NM (43%), and southern YTR (32%). P forced for southern PR with an area per region of 23%.

Decreased energy limitation in PR was characterized by a dominance of decreasing precipitation. The spatial pattern of dominant factors underlined the relevance of climate and ecosystem variables in inducing shifts in Corr(ET, VPD)-Corr(ET, SM).

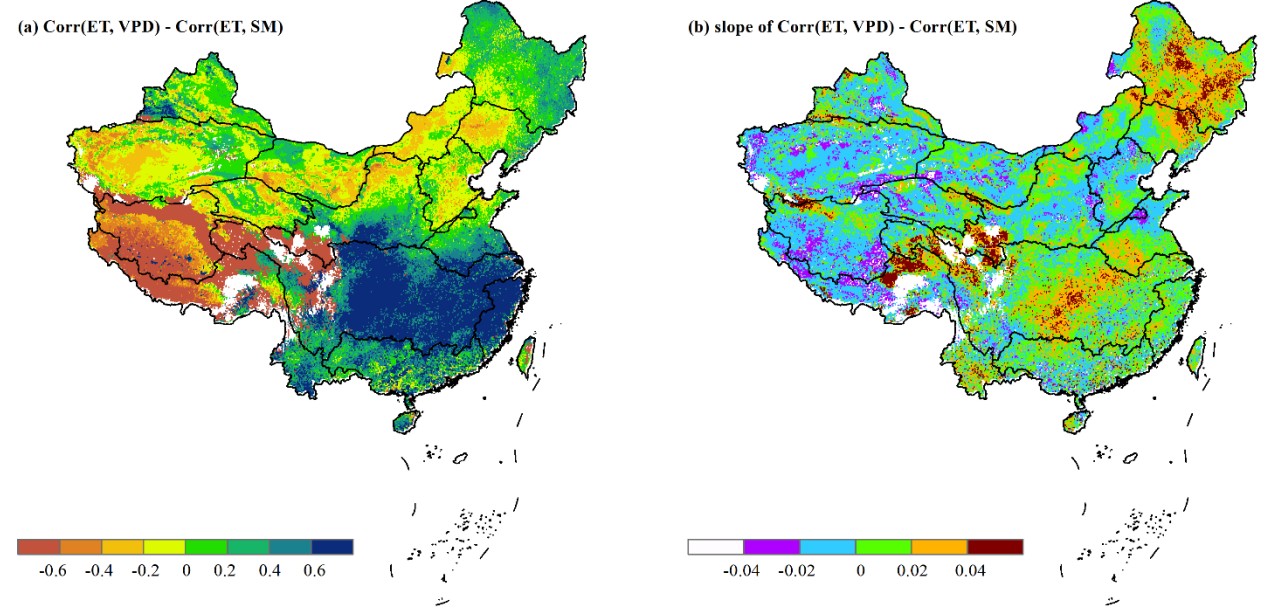

**Figure 7: Spatial pattern of (a) Corr(ET, VPD)-Corr(ET, SM) in June–September over the period 2001–2018 from PML ET and**
**10 cm soil depth soil moisture and (b) its slope using the Mann–Kendall test (Mann, 1945; Kendall, 1948).**





**Figure 8: (a)** Attribution of Corr(ET, VPD)-Corr(ET, SM) variations in June–September over the period 2001–2018 from PML to land-atmosphere variables including incoming shortwave radiation (Rs), PML evapotranspiration (ET), precipitation (P), GOSIF gross primary production (GPP), and soil moisture (SM). Colors indicate the variables that best predict the Corr(ET, VPD)-Corr(ET, SM) dynamics calculated based on the ridge regressive coefficient and the slope of normalized independent factor. Slope





**of normalized (b) Rs, (c) ET, (d) P, (e) GPP, and (f) SM in June–September over the period 2001–2018 using the Mann–Kendall test (Mann, 1945; Kendall, 1948).**

## 4 Discussion

Multiple factors contributed to the inherent constraints and potential opportunities associated with the utilization of multi-
source satellite-based water and carbon fluxes: (1) ET and GPP exhibited great uncertainties (Liu et al., 2021) in areas with dense vegetation (i.e., EBF) and barren land as indicated in Section 3.1; (2) the performance of correlation-difference method (Denissen et al., 2020) was influenced by the energy variables (VPD, Ta, and Rs) as demonstrated in Section 3.2; (3) SM from ground samplings and gridded sources (Koster et al., 2009) contributed to the uncertainty in characterizing CSM as discussed in Section 3.3; (4) by considering variations of energy and water limitations in terrestrial ecosystems (Section 3.4),
there is potential to improve the water and carbon fluxes estimation in turn. To address the question of how constraints imposed by water supply and availability of energy are defined by soil textures and plant features, there has been a growing focus on CSM from site to continental scales. For instance, Northern California exhibits CSM of 0.15 m$^3$/m$^3$ in semi-arid grassland at the site scale (Baldocchi et al., 2004); CSM using satellite-based surface temperature diurnal amplitude in semi-arid grassland of Africa has been reported to be 0.12 m$^3$/m$^3$ at the continental scale (Feldman et al. 2019). Locations
characterized by high humidity, such as tropical West Africa and the southern part of the Congo Basin (Feldman et al. 2022), exhibit high CSM. The analysis of spatial patterns of CSM using multi-source satellite-based water and carbon fluxes (Fig. 2) derived from different methods (Fig. 3) further enables us to effectively reflect the variations from energy to water limitation in spatiotemporal continuous grid cells.

The representation of the plant-accessible SM has been considered at four layers. The interlayer differences are mainly
reflected in TR and HAR as shown in Fig. 4. Researchers have found that plant species around the world have different water use depths (Feldman et al., 2023). Deep-rooted forests can better regulate their response to drought with stable CSM among soil layers. Plants exhibit a great vertical water uptake range to cope with seasonal or severe water scarcity, with water uptake often extending to well below 50 cm (Case et al., 2020), even extending to 1–2 m (Tumber-Davila et al., 2022). This means that root systems of plants play a key part in determining water and energy-limited regimes. In addition, our
CSM detection shows that fine soils and luxuriant vegetation have wetter CSM as the result shown in Fig. 5. Teuling et al. (2009) measured large-scale CSM ranging from 0.167 m$^3$/m$^3$ for sandy soil to 0.424 m$^3$/m$^3$ for clay soil using soil parameters developed by German Weather Service. CSM depends on roots pulling water out of the unsaturated soil matrix (Feldman et al., 2019). Surfaces characterized by a higher degree of plant coverage mitigate the impact of surface environmental changes to a larger extent compared to bare soil. This is due to the ability of plants to alleviate the impacts of water stress via their
stomata and root systems (Gallego-Elvira et al., 2016). For specific plants, CSM is around 0.24 m$^3$/m$^3$ in NM (Fig. 6) characterized mostly by an abundance of grass coverage. That is in line with the grassland CSM of 0.214 m$^3$/m$^3$ from the VPD-GPP-SM approach across 195 global sites from Integrated Carbon Observation System, AmeriFlux, and the FLUXNET2015 (Fu et al. 2022b). Another study based on the correlation-difference method using SM from the European





Space Agency Climate Change Initiative program and ET from FLUXCOM reports large-scale CSM of around 0.21 m³/m³
throughout Europe across all grid cells (Denissen et al., 2020). It may help understand regional or continental-scale water
and energy-limited regimes that arise from different vegetation and soil conditions.

The transition between different regimes is influenced by variables unrelated to moisture availability, such as topography. It
was interesting that although the PR was an energy-limited area (Fig. 7a), its energy limitation decreased (Fig. 7b) and
precipitation was one of the main control factors (Fig. 8), which might be related to the Karst landform (Liu, H. et al. 2020)
and needed further analysis. It should be noted that the process of deforestation and subsequent conversion to savanna would
result in significant modifications to the inherent characteristics of the vegetation and soil, which are fundamental
components of the concept known as CSM. Besides, considering the non-linear relationship between the surface energy
balance and the surface SM state (Denissen et al., 2020; Feldman et al. 2022), CSM would respond differently to external
forcings in water and energy-limited regimes. The non-linear impact also seems to be responsible for a shift in areas of
China towards more interconnected climate regimes (Zhang et al., 2020). Water-limited regions exhibit great sensitivity in
hydrologic cycles to variations in vegetation functioning, climate variability, and catchment physical conditions.
Consequently, water-limited vegetation exhibits a higher degree of sensitivity to surface disturbances compared to locations
with higher levels of precipitation. In this scenario, the effects of ET are more pronounced, resulting in a decline in energy
limitation (Fig. 8). To comprehend the underlying factors driving CSM, it is necessary to do a more comprehensive analysis
of climate and ecosystem conditions. On the one hand, the presence of wet surfaces has the potential to mitigate the impact
of radiation on the energy balance. In energy-limited regimes, energy limitation is only determined by the need for reduced
sensitivity to evaporative processes (Seneviratne et al., 2010). On the other hand, the idea that heightened responsiveness
may be attributed to the lower stomatal resistance and shorter rooting systems in water-limited environments was
demonstrated using AMSR-E satellite data (Konings and Gentine, 2017). The mechanism identified here seems to be one of
the reasons why different regions have different responses to climate change.

## 5 Conclusion

Our main accomplishment is observing and identifying water and energy limit shifts using multi-source satellite-based water
and carbon fluxes over China. These shifts show which areas are more likely to be affected by climate change. To do so, we
first examined the consistency of ET and GPP derived from the site and satellite-based grid observations and the consistency
of CSM derived from the EF-SM, VPD-GPP-SM, and correlation-difference methods. Then, satellite-based CSM from the
four ET products, the four GPP products, and the latest SM dataset was estimated and evaluated. Based on the spatial pattern
of CSM, we further quantified CSM among land cover types, soil textures, and water resource subregions and attributed the
dominant factors of water and energy limit shifts.

We discovered that CSM detected by the covariance between VPD and GPP and CSM from the correlation-difference metric
using VPD, ET, and SM match well with CSM from the EF-SM method at the site scale, suggesting that these methods

could be used to detect large-scale CSM. Surface water and energy-limited regimes varied among land cover types, soil textures, and water resources subregions. Soil texture of clay, land cover types of forests, and water resources subregions of PR and SER (south and southeast China) had high CSM; Precipitation showed a more dominant position in energy-limited southern PR. While sand content higher than 90% and barren located in western areas had the lowest CSM in the water-limited stage; ET was the dominant factor for regime shifts in water-limited western and northern regions. The 18 years of SM data were quite typical of the long-term climatology of continental wetness. Applying our analysis to CSM has considerable significance in the evaluation of global climate change impacts on regional terrestrial ecosystems over extended periods.

## Data availability

All datasets used in this study are publicly available from the referenced sources.

## Author contributions

Yi Liu: investigation, methodology, formal analysis, conceptualization, writing (original draft and review and editing); Jingfeng Xiao: supervision, writing (review and editing); Xing Li: writing (review and editing); Yue Li: writing (review and editing).

## Competing interests

The contact author has declared that none of the authors has any competing interests

## Financial support

Y. Liu acknowledges support from Guangxi Natural Science Foundation under Grant No. 2023JJB150133 and Guangxi University (Grant No. A3030051071).

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
