# Peer review of "Critical soil moisture detection and water-energy limit shift attribution using satellite-based water and carbon fluxes over China"

_Hydrology and Earth System Sciences, 2024_

## Referee Comment (RC1)

This study applies multiple methods to identify critical soil moisture (CSM) that separates water- and energy-limited regimes using several satellite-based data and in-situ observations with a specific spatial scope. Then, it explores the factors that dominate the variations using a feature regularization technique. I support this study as I think their analyses on CSM and the determinant factor advance the science on the water and energy cycle over land. However, I think the readability of the paper and the description of the analyses should be further improve. Moreover, I have concerns about the confidence in the CSM estimates. I suggest a major revision. Please see my comments below:

I do not wish to remain anonymous – Hsin Hsu

**Major Comments**

1. The article is very hard to read because the amount of abbreviation is overwhelming. I had to look up what an abbreviation stands for multiple times in just one sentence as they are from different categories (variables, locations, names of in-situ sites, land cover types, algorithms, statistical parameters, data products, etc.). I suggest retaining the full names for land cover types and some algorithms in the writing, as many do not occur frequently in the paper. The authors could also consider separating the abbreviations by different systems. For example, use Greek alphabets for statistical parameters and italics for variables.

The main methodologies used in this study may need abbreviations due to their lengthy names (e.g., Corr(ET,VPD) - Corr(ET,SM)). Sometimes they occur many times in one paragraph (or even one sentence), but the key information separating different correlation-difference methods is the second variable used in the first correlation calculation. The authors could consider modifying the notation of Denissen et al. 2020 to define:
$\Delta Corr_{Var} = Corr(ET,Var) - Corr(ET,SM)$.

2. Most of the cited work on critical soil moisture and regime examination is published before 2022. There are many new aspects of regimes and CSM since 2023. Not required to reference but the authors could consider integrating these recent studies:

i. New method for calculating CSM based on satellite data:
   *Fu et al. (2024). Global critical soil moisture thresholds of plant water stress.* *https://doi.org/10.1038/s41467-024-49244-7*
ii. Global estimation of CSM based on soil moisture dry-down framework and an index to quantify vulnerability:
   *Dong et al. (2023). Land Surfaces at the Tipping-Point for Water and Energy Balance Coupling.* *https://doi.org/10.1029/2022WR032472*
iii. In the abstract (line 11), author mentions that regimes can shift under climate change. This is not discussed in the introduction:
   *Hsu, H., Dirmeyer, P.A. (2023). Soil moisture-evaporation coupling shifts into new gears under increasing CO2.* *https://doi.org/10.1038/s41467-023-36794-5*
   *Hsu, H., Dirmeyer, P.A. (2023). Uncertainty in Projected Critical Soil Moisture Values in CMIP6 Affects the Interpretation of a More Moisture-Limited World.* *https://doi.org/10.1029/2023EF003511*
   *Duan et al. (2023). Coherent Mechanistic Patterns of Tropical Land Hydroclimate Changes.* *https://doi.org/10.1029/2022GL102285*.

3. The robustness of each technique for estimating CSM is not described:

i.    From Figure 4, it seems that CSM can be identified in almost all grid cells by the covariance method. Is this also true for all other methods? If not, what is the rate of agreement on the existence of CSM among all the methods?

ii.   Do all methods for estimating CSM have significant tests? For example, when using the SM-EF method, it is common to use three different linear models (flat line, a positive slope line, and a linear-plus-plateau) and select the best one based on the Bayesian information criterion (BIC). If flat-line regression or positive-slope regression outperforms linear-plus-plateau regression, CSM should be considered as not identified. This procedure is also used for detecting CSM by the soil moisture-drydown framework.

iii.  When using the correlation-difference method, if there is more than one SM value where the correlation-difference is zero, which SM value is identified as CSM? In Figure 3b, the red line locates at the wetter SM value when the correlation-difference is zero, but blue-green line locates at the drier SM value when the correlation-difference is zero. This should be clarified. Additionally, I assume that if either correlation is not statistically significant when calculating the correlation before taking the difference, CSM should also be treated as not identified. Is this the case in this study?

4. When examining the alignment of CSM between different methods, statistical significance is needed. I recommend a Chi-square test as it can address scenarios involving categorical data: comparing rates or proportions between two groups when the outcome is a binary variable, such as negative and positive outcomes. In this specific case, categorical data represent soil moisture (SM) values tagged as "drier-than-CSM" and "wetter-than-CSM". So, a Chi-square test can be used to compare the proportions of SM values below and above CSM between two sets of variables or groups (obtained by different methods). If there are significant differences, it means the CSM is different.

5. Figure 5 seems problematic. The CSM is extremely well aligned among different ET products. However, I assume the spread in temporal variation among ET products over many locations based on Figure 2a, where the correlation of each product's ET to in-situ data can be very different. Does that not lead to a different CSM estimate? The authors could provide some supporting information to justify the consensus, which looks too good. The tick labels on the y-axis in each bar chart are incorrect. The bars are spatial means, so error bars should also be provided.

6. In Figure 6, the CSM among different SM layers is also extremely well aligned (if I interpret it correctly). This makes me doubt the reliability of SM in deeper layers. I assume SM-EF could be decoupled in deeper soils if roots do not reach that deep in

some places, so there should be some inconsistency in CSM values. In either case, I suggest the authors provide additional analysis to examine soil moisture at deeper layers (maybe taking some grid cells for example and put as supporting information) and discuss the uncertainty of using these data products in the discussion. For example, some ET products are estimated; are the sources of input to derive ET independent of each other for all of them? Does method to derive SM at different layer inherently lead to consistency in CSM?

7. How does the author determine the set of input features for the ridge regression and why air temperature and VPD are not considered in the analysis?

**Minor Comments**

1. Line 213: should be "ea" not "ea."
2. Line 122: The term REddyProc is not explained nor mentioned elsewhere.
3. I suggest putting the unit of variables in every chart.
4. Is $\Delta$Corr calculated monthly between June and September and then an annual mean is obtained?
5. The word "Slope" in figures 7 and 8 is confusing. Is this a temporal trend? What is the statistical significance?
6. Does the author perform cross-validation or bootstrapping for ridge regression?

---

## Author Response (AR1)

Dear Dr. Adriaan J. (Ryan) Teuling,

Thank you for your letter and comments concerning our manuscript (No. hess-2024-105). We studied comments carefully, and have made corrections that we hope meet with approval. Reviewers' comments are shown in grey shadow; authors' reply is shown in regular text and all the changes are highlighted in the tracked-changes version. The main corrections in the paper and the responses to the reviewer's comments are as follows:

...............................................................

The manuscript has now been seen by two external referees. Both see considerable potential in your work, but also identify a number of issues that should be addressed before the manuscript can be considered for final publication in HESS. Most of these issues are relatively minor in nature. I therefor believe that revisions are necessary. Please prepare a revised version of your manuscript that accounts for all comments by the referees. In addition, submit a full rebuttal letter that addresses all issues raised during the review. For this, you can simply copy-paste the responses you already prepared when appropriate. I am looking forward to receiving your revised manuscript.

...............................................................

**Response to Reviewer #1:**

This study applies multiple methods to identify critical soil moisture (CSM) that separates water- and energy-limited regimes using several satellite-based data and in-situ observations with a specific spatial scope. Then, it explores the factors that dominate the variations using a feature regularization technique. I support this study as I think their analyses on CSM and the determinant factor advance the science on the water and energy cycle over land. However, I think the readability of the paper and the description of the analyses should be further improved. Moreover, I have concerns about the confidence in the CSM estimates. I suggest a major revision. Please see my comments below:
I do not wish to remain anonymous – Hsin Hsu

**Response:** We thank Dr. Hsu for the positive evaluation and constructive comments. We have carefully revised the manuscript to address the concerns and to improve the paper. Specifically, we are very grateful to Dr. Hsu for asking professional questions and providing the latest research literature. Based on these comments, we have filled the statistical knowledge gaps of critical soil moisture (CSM), such as the application of the BIC and Chi-square test, and so on. For Figure 3, we first added the BIC method to evaluate the performance of three different linear models and further constrained the occurrence of multiple critical values. For Figure 4, the chi-square test was then introduced to test differences between models. In addition, the previous figures were well aligned among different products and soil layers, so we improved them to ensure that each figure showed clear differences and could provide a clearer conclusion. For the results of CSM, we added the following constraint: "For each grid cell and the entire period, negative metrics are displayed when SM is less than CSM, and positive metrics are shown when SM is greater than CSM. If there is more than one value where SM shifts between positive and negative metrics, CSM is treated as unidentified." Further, the performances of the regression model were assessed by five-fold cross-validation using the mean absolute percentage error. Temperature and VPD were considered in the attribution analysis. The comments are shown in grey shadow; our reply is shown in regular text and all the changes are highlighted. The main responses are as follows:

Major Comments

1. The article is very hard to read because the amount of **abbreviation is overwhelming**. I had to look up what an abbreviation stands for **multiple times** in just one sentence as they are from different categories (variables, locations, names of in-situ sites, land cover types, algorithms, statistical parameters, data products, etc.). I suggest **retaining the full names** for land cover types and some algorithms in the writing, as many do not occur frequently in the paper. The authors could also consider separating the abbreviations by different systems. For example, use Greek alphabets for statistical parameters and italics for variables.

The main methodologies used in this study may need abbreviations due to their lengthy names (e.g., Corr(ET,VPD) - Corr(ET,SM)). Sometimes they occur many times in one paragraph (or even one

sentence), but the key information separating different correlation-difference methods is the second variable used in the first correlation calculation. The authors could consider **modifying the notation** of Denissen et al. 2020 to define:

ΔCorrVar = Corr(ET,Var) - Corr(ET,SM).

**Response:** Thanks for your suggestion. In the revision, we have substantially reduced the usage of abbreviations. We chose to retain the abbreviations of the eight satellite-based products, six main variables (ET, GPP, VPD, SM, EF, and CorrVPD), and site names. Acronyms for temperature and radiation in the relevant Correlation-difference method were deleted. Thus, we only kept CorrVPD = Corr(ET,VPD) - Corr(ET,SM). In addition, the full names of land cover types and sub-basins were used and the occurrence of land cover types and sub-basins was also reduced.

2. Most of the cited work on critical soil moisture and regime examination is published before 2022. There are many new aspects of regimes and CSM since 2023. Not required to reference but the authors could consider integrating these recent studies:

New method for calculating CSM based on satellite data:

Fu et al. (2024). Global critical soil moisture thresholds of plant water stress. https://doi.org/10.1038/s41467-024-49244-7

Global estimation of CSM based on soil moisture dry-down framework and an index to quantify vulnerability:

Dong et al. (2023). Land Surfaces at the Tipping-Point for Water and Energy Balance Coupling. https://doi.org/10.1029/2022WR032472

In the abstract (line 11), author mentions that regimes can shift under climate change. This is not discussed in the introduction:

Hsu, H., Dirmeyer, P.A. (2023). Soil moisture-evaporation coupling shifts into new gears under increasing CO2. https://doi.org/10.1038/s41467-023-36794-5

Hsu, H., Dirmeyer, P.A. (2023). Uncertainty in Projected Critical Soil Moisture Values in CMIP6 Affects the Interpretation of a More Moisture-Limited World. https://doi.org/10.1029/2023EF003511

Duan et al. (2023). Coherent Mechanistic Patterns of Tropical Land Hydroclimate Changes. https://doi.org/10.1029/2022GL102285.

**Response:** Thanks for your recommendation. We have added these studies in the Introduction and Methods section as follows:

Lines 48–54: "Traditionally, under the framework based on the ratio of LE to the total of LE and H (Haghighi et al., 2018; Fu et al., 2022b), sparse eddy covariance observations (Feldman et al., 2019; Fu et al., 2022a) pose challenges in adequately capturing comprehensive regional or continental-scale CSM and its variations (Dong et al., 2023; Hsu and Dirmeyer, 2023a). In recent years, the feasibility of conducting large-scale analysis has been enhanced by the growing accessibility of multi-source satellite-based datasets (Liu et al., 2012). Globally, some model-based analyses used the ratio of LE to net radiation (Seneviratne et al., 2010; Schwingshackl et al., 2017), surface temperature diurnal amplitude (Feldman et al., 2019; Fu et al., 2024), and LE (Hsu and Dirmeyer, 2023b; Duan et al., 2023)."

Lines 200–201: "The alignment of CSM obtained by different methods was determined using the chi-square test (McHugh, 2013; Hsu and Dirmeyer, 2023a)."

3. The robustness of each technique for estimating CSM is not described:

From **Figure 4**, it seems that CSM can be identified in almost all grid cells by the covariance method. **Is this also true for all other methods?** If not, what is the **rate of agreement** on the existence of CSM among all the methods?

**Response:** No, CSM cannot be identified in all grid cells. Thanks for reminding me. In the previous version, we did an average for each tipping point in each year and treated it as a critical value, which will reduce the difference between different products. In the revised version, we considered that only one tipping point in each year is the critical value.

We added the constraints in Method:

Lines 155–157: "For each grid cell and the entire period per year, negative metrics are displayed when

SM is less than CSM, and positive metrics are shown when SM is greater than CSM. If there is more than one value where SM shifts between positive and negative metrics, CSM is treated as unidentified."

We added the disparity of soil moisture regime and percentage of area with p < 0.05 in Result:

Lines 265–267: "The number of wet binary bit was used to quantify the agreement among eight ET and GPP-based models at 10 cm soil depth. If CSM was identified, the SM wetter than CSM was represented as 1, and 0 for others. If CSM was not identified within a year, digits of the mode were treated as 0. If CSM was not detected for all 18 years, it was displayed as empty."

For Figure 4, the agreement and chi-square test were introduced to test the differences between the models:

[Figure]

1: CAMELE ET and GLASS ET
2: CAMELE ET and PML ET
3: CAMELE ET and SEBAL ET
4: CAMELE ET and GLASS GPP
5: CAMELE ET and GOSIF GPP
6: CAMELE ET and PML GPP
7: CAMELE ET and TL GPP
8: GLASS ET and PML ET
9: GLASS ET and SEBAL ET
10: GLASS ET and GLASS GPP
11: GLASS ET and GOSIF GPP
12: GLASS ET and PML GPP
13: GLASS ET and TL GPP
14: PML ET and SEBAL ET
15: PML ET and GLASS GPP
16: PML ET and GOSIF GPP
17: PML ET and PML GPP
18: PML ET and TL GPP
19: SEBAL ET and GLASS GPP
20: SEBAL ET and GOSIF GPP
21: SEBAL ET and PML GPP
22: SEBAL ET and TL GPP
23: GLASS GPP and GOSIF GPP
24: GLASS GPP and PML GPP
25: GLASS GPP and TL GPP
26: GOSIF GPP and PML GPP
27: GOSIF GPP and TL GPP
28: PML GPP and TL GPP

**Figure 4: Spatial pattern of number of wet binary bit at 10 cm depth using covariance between vapor pressure deficit (VPD) and gross primary production (GPP) from (a) GOSIF, (b) GLASS, (c) PML, (d) TL. Spatial pattern of number of wet binary bit at 10 cm depth using correlation-difference metric with Kendall's rank correlation (Corr) between detrended anomaly of soil moisture (SM) and evapotranspiration (ET) from (e) CAMELE, (f) GLASS, (g) PML, and (h) SEBAL and Corr between detrended anomaly of VPD and those ET products. (i) Disparity of soil moisture regimes among all methods and (j) the percentage of area with p < 0.05. ZGE: Zhungaer Basin, PR: Pearl River Basin, YTR: Yangtze River Basin, SWR: Southwestern River Basin, TR: Tarim Basin, SR: Songhua River Basin, CT: Changthang Region, NM: Inner Mongolian Plateau Region, LR: Liaohe River Basin, YR: Yellow River Basin, HR: Huaihe River Basin, HC: Hexi Corridor Region, HAR: Haihe River Basin, SER: Southeastern River Basin, QB: Qaidam Basin.**

Do all methods for estimating CSM have significant tests? For example, when using the SM-EF method, it is common to use three different linear models (flat line, a positive slope line, and a linear-plus-plateau) and select the best one based on the **Bayesian information criterion (BIC)**. If flat-line regression or positive-slope regression outperforms linear-plus-plateau regression, CSM should be considered as not identified. This procedure is also used for detecting CSM by the soil moisture-drydown framework.

**Response:** Thanks for the question. No, we didn't have significant tests in the previous version. Thus, for Figure 3, we have added the BIC method to evaluate the performance of three different linear models and further constrained the occurrence of multiple critical values.

We added the description in Method:

Lines 174–177: "In addition, the Bayesian Information Criterion (BIC) (Schwarz 1978) was used to select the best fit among three-segmented regression candidates (the flat line, the positive slope line, and the linear-plus-plateau). If the flat-line regression or the positive-slope regression outperformed the linear-plus-plateau regression, CSM was considered as not identified."

We added the description in Results:

Lines 240–244: "Variations of SM and EF were depicted in Figure 3 for eight sites. Fitted lines represented controlling mechanisms in various evaporative regimes. Overall, the linear-plus-plateau regression with the lowest BIC outperformed the flat line and the positive slope line in the study period of all eight sites. Specifically, CN-Du2 and Qianyanzhou sites showed a great slope at low SM values with BIC of -80.29 and -98.64, respectively."

[Figure]

**Figure 3: Variations of soil moisture and evaporative fraction at (a) Xilingela in 2004, (c) Damshung in 2004, (e) CN-Sw2 in 2011, (g) CN-Du2 in 2007, (i) CN-Cng in 2010, (k) Miyun in 2009, (m) Huailai in 2015, (o) Qianyanzhou in 2010. Variations of covariance (referred to as Cov) between vapor pressure deficit and gross primary production, and correlation-difference metric (referred to as CorrVPD) at (b) Xilingela in 2004, (d) Damshung in 2004, (f) CN-Sw2 in 2011, (h) CN-Du2 in 2007, (j) CN-Cng in 2010, (l) Miyun in 2009, (n) Huailai in 2015, (p) Qianyanzhou in 2010.**

When using the correlation-difference method, if there is more than one SM value where the correlation-difference is zero, which SM value is identified as CSM? In **Figure 3b**, the red line locates at the wetter SM value when the correlation-difference is zero, but blue-greenline locates at the drier SM value when the correlation-difference is zero. This should be clarified.

**Response:** In the revised version, we consider that only one tipping point in each year is the critical value. We added the constraints in Method:

Lines 155–157: "For each grid cell and the entire period per year, negative metrics are displayed when SM is less than CSM, and positive metrics are shown when SM is greater than CSM. If there is more than one value where SM shifts between positive and negative metrics, CSM is treated as unidentified."

We added the description in Result:

Lines 250–257: "For CN-Du2 and Qianyanzhou sites, only positive or negative Cov and CorrVPD were found. For Damshung, CN-Cng, and Huailai sites, we found more than one SM value where the Cov or CorrVPD was zero. Along with surface soil wetting, there was a change of Cov and CorrVPD from positive to negative at these sites, inconsistent with the transition from water to energy limitation, indicating that CSM was not identifiable. Different from above, Cov had the optimal CSM value that agreed best with the EF-SM-derived CSM at Xilingela and CN-Sw2 sites. Through another technique, CorrVPD was better than Cov at Miyun site. In these sites, Cov and CorrVPD changed from negative (water limit) to positive (energy limit). Therefore, CorrVPD and Cov had the potential to obtain large-scale CSM."

Additionally, I assume that if either correlation is **not statistically significant** when calculating the correlation before taking the difference, CSM should also be treated as not identified. Is this the case in this study?

**Response:** No, since each set of correlations has only 9 values, satisfying a significant correlation at all times is hard to achieve. Thus, although no significant correlation is verified, we added the following constraint:

Lines 155–157: "For each grid cell and the entire period per year, negative metrics are displayed when SM is less than CSM, and positive metrics are shown when SM is greater than CSM. If there is more than one value where SM shifts between positive and negative metrics, CSM is treated as unidentified."

Lines 248–250: "To explore the performance of both methods on sites and whether they can be used on a large scale, the data applied to both methods was averaged for 8 days, consistent with gridded data with the 8-day time scale."

4. When examining the alignment of CSM between different methods, **statistical significance** is needed. I recommend a **Chi-square test** as it can address scenarios involving categorical data: comparing rates or proportions between two groups when the outcome is a binary variable, such as negative and positive outcomes. In this specific case, categorical data represent soil moisture (SM) values tagged as "drier-than-CSM" and "wetter-than-CSM". So, a Chi-square test can be used to compare the proportions of SM values below and above CSM between two sets of variables or groups (obtained by different methods). If there are significant differences, it means the CSM is different.

**Response:** Thanks for your suggestion. We have added statistical significance in Figure 4 shown in the previous question and added the chi-square test to the Method and description of the Results.

We added the description in Method:

Lines 200–204: "The alignment of CSM obtained by different methods was determined using the chi-square test (McHugh, 2013; Hsu and Dirmeyer, 2023a). CROSSTAB in MATLAB was used to perform the chi-square test. SM values were divided into two groups, below and above CSM. In this case, categorical data was tagged as a binary variable of 0 for drier than CSM and 1 for wetter than CSM. If there were significant differences with a 95% confidence level, CSM was different."

We added the description in Result:

Lines 269–274: "Figure 4 shows the strong disparity in North and Central China, especially in Inner Mongolian Plateau Region, Songhua River Basin, Yangtze River Basin, and Yellow River Basin. In these regions, the chi-square test showed significant differences among GPP-based models due to their large number of wet binary bits. In addition, TL GPP displayed no CSM value in Northwest China. Note that the SM wetter-than-CSM showed agreement in eastern and southern basins, such as Huaihe River Basin, Liaohe River Basin, Southeastern River Basin, and Pearl River Basin, indicating that ET and GPP-based

5. Figure 5 seems problematic. The CSM is extremely well aligned among different ET products. However, I assume the spread in temporal variation among ET products over many locations based on Figure 2a, where the correlation of each product's ET to in-situ data can be very different. Does that not lead to a different CSM estimate? The authors could provide some supporting information to justify the **consensus**, which looks too good. The tick labels on the y-axis in each bar chart are incorrect. The bars are spatial means, so **error bars** should also be provided.

**Response:** Thanks for the question. Yes, products lead to a different CSM. This alignment comes from two reasons:

(1) the first is that we have done an average for each tipping point in each year and treated it as a critical value, which will reduce the difference between different products. In the revised version, we consider that only one tipping point in each year is the critical value. We added the constraints in Method "2.2 Determination of CSM":

Lines 155–157: "For each grid cell and the entire period per year, negative metrics are displayed when SM is less than CSM, and positive metrics are shown when SM is greater than CSM. If there is more than one value where SM shifts between positive and negative metrics, CSM is treated as unidentified."

In the response to question 3, we can see that in the process of SM from low to high, if there are multiple times that metrics (Cov and CorrVPD) change from negative to positive, it means that this CSM does not conform to the change process of water stress to energy stress caused by SM from low to high.

We can see that the critical values obtained for the eight products in Figures 4 and 5 also show varying undetected areas.

(2) then, the spatial variation in the study area is too great, and the CSM and SM in spatial and depth are represented by a gradual color, resulting in the depth difference is not obvious compared to the spatial difference. The color of the previous drawing can only reflect the difference in space, not the difference in depth. Therefore, we averaged the CSM and SM by land cover and soil texture, highlighting differences in SM and CSM at different depths for each type.

After the constraint was added in question 3, we modified the code, redrawn Figure 5 and added the error bars, and redrawn Figure 6.

[Figure]

**Figure 5: The spatial pattern of critical soil moisture (CSM) at 10 cm depth using covariance between vapor pressure deficit (VPD) and gross primary production (GPP) from (a) GOSIF, (b) GLASS, (c) PML, (d) TL and CSM using correlation-difference metric with Kendall's rank correlation (Corr) between detrended anomaly soil moisture (SM) and evapotranspiration (ET) from (e) CAMELE, (f) GLASS, (g) PML, and (h) SEBAL and Corr between detrended anomaly VPD and those ET products. And (i–w) the basin-average values of ZGE: Zhungaer Basin, PR: Pearl River Basin, YTR: Yangtze River Basin, SWR: Southwestern River Basin, TR: Tarim Basin, SR: Songhua River Basin, CT: Changthang Region, NM: Inner Mongolian Plateau Region, LR: Liaohe River Basin, YR: Yellow River Basin, HR: Huaihe River Basin, HC: Hexi Corridor Region, HAR: Haihe River Basin, SER: Southeastern River Basin, QB: Qaidam Basin.**

[Figure]

**Figure 6: Soil moisture (SM) at 10 cm, 20 cm, 80 cm, and 100 cm soil depths, and critical soil moisture (CSM) derived from CAMELE, GLASS, PML, and SEBAL ET at corresponding soil depths for (a) cropland, (b)**

**forests, (c) grassland, (d) water, (e) ice, (f) urban, (g) barren, soils with a majority of (h) clay, (i) silt, and sand with content (j) less than 60, (k) between 60% and 70%, (l) between 70% and 80%, (m) between 80% and 90%, and (n) higher than 90%.**

6. In Figure 6, the CSM among different SM layers is also **extremely well aligned** (if I interpret it correctly). This makes me doubt the reliability of SM in deeper layers. I assume SM-EF could be decoupled in deeper soils if roots do not reach that deep in some places, so there should be some inconsistency in CSM values.

**Response:** Thanks for the question. We improved the representation of Figure 6 to make the differences between the different depths more noticeable and changed the corresponding description as follows.

Lines 286–297: "Furthermore, large-scale CSM depended on roots pulling water out of the unsaturated soil matrix (Feldman et al., 2019) and varied across vegetation types and soil textures at four soil layers (Figure 6). With shorter root systems and less vegetation (i.e., barren), areas with low CSM were water-limited. Forest regions displayed a relatively high CSM (e.g., 0.18 m3/m3 using PML ET and 10 cm depth SM). As for soil textures, sand covering the large area was further part into content of less than 60%, 60–70%, 70–80%, 80–90%, and higher than 90%. Soil with a majority of clay had a wetter CSM than others (e.g., 0.38 m3/m3 using PML ET and 10 cm depth SM) and was to be expected given that clay had a larger negative matric potential compared to coarse soil textures dominated by sand and silt. In summary, fine soils and luxuriant vegetation had wetter CSM. Additionally, a layer-wise CSM analysis was conducted to highlight variations in SM properties for different soil layers. It was evident that there were variations in the CSM behavior across layers with higher SM and CSM at 20 cm soil depth than at other depths. We also found that there was higher CSM than SM at all four layers for grassland and clay, which identified a large range of SM within water-limited regimes. However, for cropland and forests, differences existed in CSM among four ET-based methods, with higher CSM from GLASS and SEBAL than others."

In either case, I suggest the authors provide **additional analysis** to examine soil moisture at deeper layers (maybe taking some grid cells for example and put as supporting information) **and discuss** the uncertainty of using these data products in the discussion.

**Response:** According to the previous version, the results were too consistent, and additional analysis was required to demonstrate the differences between the different depths. However, according to our latest version, the differences in different depths have been shown by land cover types and soil textures as shown in Figure 6 in question 5. Therefore, instead of providing additional analysis, we have modified the original representation as shown in Figure 6.

Because this SM dataset is based on site observation data, the uncertainty of SM comes from spatial interpolation, and the interpolation factors affecting different depths are different. Here we add a description to the Section Discussion:

We discussed the uncertainty of four soil layers in Discussion:

Lines 392–394: "For gridded SM, surface climate shows a significant effect on the upper soil layer SM modeling, while the background aridity leads to low variability of the deeper layer SM (Li, Q. et al., 2022)."

For example, some ET products are estimated; are the sources of input to derive ET independent of each other for all of them?

**Response:** No, CAMELE ET combined PML ET. We discussed the sources of input to derive ET in Discussion:

Lines 385–389: "Multiple factors contributed to inherent constraints in identifying different regimes associated with the utilization of multi-source satellite-based ET and GPP. For example, ET and GPP exhibited great uncertainties (Liu et al., 2021) in areas with barren land as indicated in Section 3.1. In eastern and southern regions (Figure 4), where satellite-based methods were more reliable, eight satellite-based SM regimes were in good agreement. Since the CAMELE ET combined PML ET, they showed consistency in cropland and forests with a lower CSM than GLASS and SEBAL (Section 3.3)."

Does method to derive SM at different layer inherently lead to consistency in CSM?

**Response:** No, the CSM obtained from different layer SM is definitely different. However, the spatial variation in the study area is too great, and the CSM and SM in spatial and depth are represented by a gradual color, resulting in the depth difference is not obvious compared to the spatial difference. The

color of the previous drawing can only reflect the difference in space, not the difference in depth. Therefore, we averaged the CSM and SM by land cover and soil texture, highlighting differences in SM and CSM at different depths for each type as shown in Figure 6 in question 5.

7. How does the author determine the set of input features for the ridge regression and why air temperature and VPD are not considered in the analysis?

**Response:** Thanks for the question. Given the similar information among radiation, temperature, and VPD, only radiation represented the atmospheric energy drive. In the revision, we have changed ridge regression to partial least square regression, added cross-validation as shown in Figure 7, and added temperature and VPD in partial least square regression as shown in Figure 8.

[Figure]

**Figure 7: Spatial pattern of (a) CorrVPD derived from PML ET and 10 cm soil depth soil moisture and (b) the mean absolute percentage error based on partial least square regression for CorrVPD estimations. ZGE: Zhungaer Basin, PR: Pearl River Basin, YTR: Yangtze River Basin, SWR: Southwestern River Basin, TR: Tarim Basin, SR: Songhua River Basin, CT: Changthang Region, NM: Inner Mongolian Plateau Region, LR: Liaohe River Basin, YR: Yellow River Basin, HR: Huaihe River Basin, HC: Hexi Corridor Region, HAR: Haihe River Basin, SER: Southeastern River Basin, QB: Qaidam Basin.**

[Figure]

**Figure 8: Spatial patterns of significance (p<0.05) of (a) CorrVPD, (b) precipitation (P), (c) incoming shortwave radiation (SRa), (d) GOSIF gross primary production (GPP), (e) PML evapotranspiration (ET), (f) soil moisture (SM), (g) temperature (Ta), and (h) vapor pressure deficit (VPD) during the period of CorrVPD detection using the Mann–Kendall test (Mann, 1945; Kendall, 1948), "De." Means "decreasing" and "In." means "increasing". (i) Attribution of CorrVPD variations. Colors indicate the variable that best predicts the CorrVPD dynamics. ZGE: Zhungaer Basin, PR: Pearl River Basin, YTR: Yangtze River Basin, SWR: Southwestern River Basin, TR: Tarim Basin, SR: Songhua River Basin, CT: Changthang Region, NM: Inner Mongolian Plateau Region, LR: Liaohe River Basin, YR: Yellow River Basin, HR: Huaihe River Basin, HC: Hexi Corridor Region, HAR: Haihe River Basin, SER: Southeastern River Basin, QB: Qaidam Basin.**

Minor Comments

1. Line 213: should be "ea" not "ea."

**Response:** We have modified it in the text.

2. Line 122: The term REddyProc is not explained nor mentioned elsewhere.

**Response:** We have added it in the text as follows:

Lines 84–88: "Given the fact that Huazhaizi, Dashalong, Luodi, Arou, Guantao, Huailai, Miyun, and Daxing did not have GPP data, REddyProc website (https://www.bgc-jena.mpg.de/5622399/REddyProc/) was used to calculate GPP. REddyProc imported half-hourly net ecosystem exchange, LE, H, and meteorological measurements to partition net ecosystem exchange into GPP and ecosystem respiration."

3. I suggest putting the unit of variables in every chart.

**Response:** We have added the unit in all figures.

4. Is ΔCorr calculated monthly between June and September and then an annual mean is obtained?

**Response:** As mentioned in our response to question 3 of Major Comments above, we got at most one critical value per year, and then an annual mean was obtained.

5. The word "Slope" in figures 7 and 8 is confusing. Is this a temporal trend? What is the statistical significance?

**Response:** Yes, it is a temporal trend. We changed the "slope" to "Sen's slope" using "the Mann–Kendall test". The significance was added in Figure 8.

6. Does the author perform cross-validation or bootstrapping for ridge regression?

**Response:** In the revision, we have changed ridge regression to partial least square regression and added cross-validation as shown in Figure 7.

**Response to Reviewer #2:**

General Comments:

This manuscript presents an analysis of critical soil moisture (CSM) across China using multiple satellite-based datasets of evapotranspiration (ET), gross primary production (GPP), and soil moisture (SM). The authors apply two methods to detect CSM - a correlation-difference approach and a VPD-GPP-SM covariance approach. They evaluate the spatial patterns of CSM across different land cover types, soil textures, and regions of China, and analyze the factors driving shifts between water and energy-limited regimes.

Overall, this study represents a substantial contribution within the scope of HESS. The use of multiple satellite datasets to examine CSM at large scales is novel and provides new insights into water and energy limitations across diverse landscapes in China. The methods are generally sound and though the results are discussed not comprehensively in the context of related work. The manuscript is in general structured, though some sections could be more concise and many textual improvements might be needed. I will give a major revision for this work.

**Response:** We thank the reviewer for the positive evaluation and constructive comments. We have carefully revised the manuscript based on your suggestions and addressed your concerns as detailed below.

Specific Comments:

1. The methods section is quite detailed, which is good for reproducibility. However, some of the **dataset descriptions are too redundant** right now and could potentially be shortened or moved to data availability statement to improve readability of the main text. For example, subsections such as 2.1.2, 2.1.4, 2.1.5.

**Response:** Thanks for your suggestion. We moved the websites of the data to the Section of Data Availability.

Below is the revised main text:

[revised manuscript text omitted]

2. The manuscript uses **too many abbreviations** from various categories (region name, physical variable, mathematical methods, etc), making it difficult to read. Consider retaining full names for less frequently used terms to improve clarity.

**Response:** Thanks for your suggestion. In the revision, we have substantially reduced the usage of abbreviations. We chose to retain the abbreviations of the eight satellite-based products, six main variables (ET, GPP, VPD, SM, EF, and CorrVPD), and site names. Acronyms for temperature and radiation in the relevant Correlation-difference method were deleted. Thus, we only kept CorrVPD = Corr(ET,VPD) - Corr(ET,SM). In addition, the full names of land cover types and sub-basins were used and the occurrence of land cover types and sub-basins was also reduced.

3. The evaluation of satellite-based ET and GPP products against flux tower data (Section 3.1) is valuable. However, **more discussion (section 4) of the implications of biases** in these products for the CSM analysis would strengthen the paper.

**Response:** Thanks for your suggestion. We have added the discussion of the implications of biases in the products in Section 4 as follows:

Lines 385–389: "Multiple factors contributed to inherent constraints in identifying different regimes associated with the utilization of multi-source satellite-based ET and GPP. For example, ET and GPP exhibited great uncertainties (Liu et al., 2021) in areas with barren land as indicated in Section 3.1. In eastern and southern regions (Figure 4), where satellite-based methods were more reliable, eight satellite-based SM regimes were in good agreement. Since the CAMELE ET combined PML ET, they showed consistency in cropland and forests with a lower CSM than GLASS and SEBAL (Section 3.3)."

4. The comparison of CSM detection methods at flux tower sites (Section 3.2) is an important component. The authors could consider adding **a quantitative metric of agreement** between methods and also a statistical test to assess the **significance of the agreement** to supplement the qualitative comparisons.

**Response:** Thanks for the suggestion. We introduced the agreement and chi-square test to test the difference and significance between the models, and modified the text as follows:

Lines 265–267: "The number of wet binary bit was used to quantify the agreement among eight ET and GPP-based models at 10 cm soil depth. If CSM was identified, the SM wetter than CSM was represented as 1, and 0 for others. If CSM was not identified within a year, digits of the mode were treated as 0. If CSM was not detected for all 18 years, it was displayed as empty."

For Figure 4, the agreement and chi-square test were introduced to test the differences between the models:

[Figure]

**Figure 4: Spatial pattern of number of wet binary bit at 10 cm depth using covariance between vapor pressure deficit (VPD) and gross primary production (GPP) from (a) GOSIF, (b) GLASS, (c) PML, (d) TL. Spatial pattern of number of wet binary bit at 10 cm depth using correlation-difference metric with Kendall's rank correlation (Corr) between detrended anomaly of soil moisture (SM) and evapotranspiration (ET) from (e) CAMELE, (f) GLASS, (g) PML, and (h) SEBAL and Corr between detrended anomaly of VPD and those ET products. (i) Disparity of soil moisture regimes among all methods and (j) the percentage of area with p < 0.05. ZGE: Zhungaer Basin, PR: Pearl River Basin, YTR: Yangtze River Basin, SWR: Southwestern River Basin, TR: Tarim Basin, SR: Songhua River Basin, CT: Changthang Region, NM: Inner Mongolian Plateau Region, LR: Liaohe River Basin, YR: Yellow River Basin, HR: Huaihe River Basin, HC: Hexi Corridor Region, HAR: Haihe River Basin, SER: Southeastern River Basin, QB: Qaidam Basin.**

5. The high level of alignment in CSM estimates among different ET products and soil moisture layers is surprising given the potential variability in these datasets.

Additional analysis or explanation is needed to **justify this consistency** and **discuss potential uncertainties**.

**Response:** Thanks for the question. This alignment comes from two reasons:

(1) the first is that we have done an average for each tipping point in each year and treated it as a critical value, which will reduce the difference between different products. In the revised version, we consider

that only one tipping point in each year is the critical value. We added the constraints in Method "2.2 Determination of CSM":

Lines 155–157: "For each grid cell and the entire period per year, negative metrics are displayed when SM is less than CSM, and positive metrics are shown when SM is greater than CSM. If there is more than one value where SM shifts between positive and negative metrics, CSM is treated as unidentified."

In the response to question 3, we can see that in the process of SM from low to high, if there are multiple times that metrics (Cov and CorrVPD) change from negative to positive, it means that this CSM does not conform to the change process of water stress to energy stress caused by SM from low to high.

We can see that the critical values obtained for the eight products in Figure 5 also show varying undetected areas.

(2) then, the spatial variation in the study area is too great, and the CSM and SM in spatial and depth are represented by a gradual color, resulting in the depth difference is not obvious compared to the spatial difference. The color of the previous drawing can only reflect the difference in space, not the difference in depth. Therefore, we averaged the CSM and SM by land cover and soil texture, highlighting differences in SM and CSM at different depths for each type.

After the constraint was added in question 3, we modified the code, redrawn Figure 5 and added the error bars, and redrawn Figure 6.

We improved the representation of Figure 6 to make the differences between the different depths more noticeable and changed the corresponding description as follows.

Lines 286–297: "Furthermore, large-scale CSM depended on roots pulling water out of the unsaturated soil matrix (Feldman et al., 2019) and varied across vegetation types and soil textures at four soil layers (Figure 6). With shorter root systems and less vegetation (i.e., barren), areas with low CSM were water-limited. Forest regions displayed a relatively high CSM (e.g., 0.18 m3/m3 using PML ET and 10 cm depth SM). As for soil textures, sand covering the large area was further part into content of less than 60%, 60–70%, 70–80%, 80–90%, and higher than 90%. Soil with a majority of clay had a wetter CSM than others (e.g., 0.38 m3/m3 using PML ET and 10 cm depth SM) and was to be expected given that clay had a larger negative matric potential compared to coarse soil textures dominated by sand and silt. In summary, fine soils and luxuriant vegetation had wetter CSM. Additionally, a layer-wise CSM analysis was conducted to highlight variations in SM properties for different soil layers. It was evident that there were variations in the CSM behavior across layers with higher SM and CSM at 20 cm soil depth than at other depths. We also found that there was higher CSM than SM at all four layers for grassland and clay, which identified a large range of SM within water-limited regimes. However, for cropland and forests, differences existed in CSM among four ET-based methods, with higher CSM from GLASS and SEBAL than others."

[Figure]

**Figure 5: The spatial pattern of critical soil moisture (CSM) at 10 cm depth using covariance between vapor pressure deficit (VPD) and gross primary production (GPP) from (a) GOSIF, (b) GLASS, (c) PML, (d) TL and CSM using correlation-difference metric with Kendall's rank correlation (Corr) between detrended anomaly soil moisture (SM) and evapotranspiration (ET) from (e) CAMELE, (f) GLASS, (g) PML, and (h) SEBAL and Corr between detrended anomaly VPD and those ET products. And (i–w) the basin-average values of ZGE: Zhungaer Basin, PR: Pearl River Basin, YTR: Yangtze River Basin, SWR: Southwestern River Basin, TR: Tarim Basin, SR: Songhua River Basin, CT: Changthang Region, NM: Inner Mongolian Plateau Region, LR: Liaohe River Basin, YR: Yellow River Basin, HR: Huaihe River Basin, HC: Hexi Corridor Region, HAR: Haihe River Basin, SER: Southeastern River Basin, QB: Qaidam Basin.**

[Figure]

**Figure 6: Soil moisture (SM) at 10 cm, 20 cm, 80 cm, and 100 cm soil depths, and critical soil moisture (CSM) derived from CAMELE, GLASS, PML, and SEBAL ET at corresponding soil depths for (a) cropland, (b)**

**forests, (c) grassland, (d) water, (e) ice, (f) urban, (g) barren, soils with a majority of (h) clay, (i) silt, and sand with content (j) less than 60, (k) between 60% and 70%, (l) between 70% and 80%, (m) between 80% and 90%, and (n) higher than 90%.**

6. Moreover, the consistency in CSM values across different soil depths raises questions about the reliability of deeper soil moisture data. Consider providing additional analysis of deep soil moisture to showcase **the possible difference** between the soil layers and discussing **the uncertainties** associated with these measurements.

**Response:** According to the previous version, the results were too consistent, and additional analysis was required to demonstrate the differences between the different depths. However, according to our latest version, the differences in different depths have been shown by land cover types and soil textures as shown in Figure 6 in question 5. Therefore, instead of providing additional analysis, we have modified the original representation as shown in Figure 6.

Because this SM dataset is based on site observation data, the uncertainty of SM comes from the spatial interpolation, and the interpolation factors affecting different depths are different. Here we add a description to the Section Discussion:

We discussed the uncertainty of four soil layers in Discussion:

Lines 392–394: "For gridded SM, surface climate shows a significant effect on the upper soil layer SM modeling, while the background aridity leads to low variability of the deeper layer SM (Li, Q. et al., 2022)."

7. The attribution analysis of water/energy limit shifts (Section 3.4) provides valuable insights. The authors could consider **expanding on the implications of these findings** for water resource management or ecosystem responses to climate change, perhaps in discussions part.

**Response:** Thanks for your suggestion. We have added the implications of these findings for water resource management or ecosystem in Section 4 as follows:

Lines 394–399: "Besides, external forcings seem to be responsible for a shift towards enhanced land-atmosphere coupling (Zhang et al., 2020). It should be noted that the South-to-North Water Diversion Project and the Pinglu Canal Project in China would result in significant modifications to SM characteristics, which are fundamental components of the concept known as CSM. Water management measures may reduce water stress in grasslands affected by climate change and make southern coastal clay areas more resistant to possible disturbances. Overall, our research could inform large-scale water conservancy projects for better allocation of water supply resources."

8. The discussion section effectively contextualizes the results within existing literature. However, it could be **strengthened by more explicitly addressing the limitations of the approach** used in this manuscript and potential future research directions.

**Response:** Thanks for your suggestion. We have added the limitations and potential opportunities in Section 4 as follows:

We added the limitations:

Lines 383–387: "However, this study focusing on a specific time of year may not be enough to explain the critical value that may be shown in the rest of the year. Since CSM values in some grids were not detected by eight products, further research is needed for the CSM that may appear in the rest of the year in different regions. In addition, to compare the performance of multi-source remotely sensed water and carbon fluxes, we unified all data into the 8-day resolution. Therefore, a more refined time scale, such as a one-day scale study, is also needed."

We added the potential opportunities:

Lines 402–404: "Future research directions could include the impact of hydraulic projects such as inter-basin water transfers on CSM, the impact of extreme disturbances such as tropical cyclones and wildfires on CSM, and possible changes in CSM."

Some textual suggestions:

1. Line 10: "suffer from water limitation", this kind of metaphor is not suitable I assume, please change it to some words that are not for humans.

**Response:** We have changed this sentence from to "Critical soil moisture (CSM), a tipping point of soil

moisture (SM) at which surface fluxes shift from energy- to water-limited regimes, is essential for the vegetation state and corresponding land‑atmosphere coupling."

2. Line 14: Put "were assessed over China" before "derived" in the sentence to let the colon directly connecting the following methods.

**Response:** We have changed this sentence as suggested.

3. Line 24: Maybe change the sentence to "Through intercomparison, CSM from multi-source ET and GPP datasets across China is found to be consistent and robust."

**Response:** We have changed this sentence as suggested.

4. Line 34: Please don't repeat sentences in your manuscript, this one is the same with that in your abstract, please revise either of them.

**Response:** We have changed this sentence to "Critical soil moisture (CSM) serves as an indicator of shifts in the relationship between water and energy availability (Schwingshackl et al., 2017; Denissen et al., 2020) and is essential in shaping regional climates."

5. Line 44: "customary" and "matric", change them to some others relatively commonly used.

**Response:** We have changed this sentence to "Diagnosing CSM across various biomes and climatic zones helps to understand water-energy limit regimes determined by distinct flora and soil types (Homae et al., 2002; Hsu and Dirmeyer, 2023b)."

6. Line 98: "comparability" is not common in papers. Some noun forms of words are not common to be used even in academic world. Please consider adjective forms and revise the relevant sentence or use other nouns. These are just what I roughly found, please read the text thoroughly after revision and also consider a text revision service.

**Response:** We have deleted this sentence and made corrections to the whole text.

7. Figure 1: Consider changing the colors of forest and grassland to make them easier to differentiate as the landcovers you have are not so many

**Response:** We have changed the colors to make it easier to differentiate forest from grassland. Below is the revised figure:

[Figure]

**Figure 1: (a) Locations of flux sites, land cover types (2020), and water resource subregions of China. Distributions of (b) clay, (c) silt, and (d) sand content (1995). ZGE: Zhungaer Basin, PR: Pearl River Basin, YTR: Yangtze River Basin, SWR: Southwestern River Basin, TR: Tarim Basin, SR: Songhua River Basin, CT: Changthang Region, NM: Inner Mongolian Plateau Region, LR: Liaohe River Basin, YR: Yellow River Basin, HR: Huaihe River Basin, HC: Hexi Corridor Region, HAR: Haihe River Basin, SER: Southeastern River Basin, QB: Qaidam Basin.**

8. Figure 5: Consider making it into two columns

**Response:** Thanks for your suggestion. We have changed the Figure 5 to:

[Figure]

**Figure 5: The spatial pattern of critical soil moisture (CSM) at 10 cm depth using covariance between vapor pressure deficit (VPD) and gross primary production (GPP) from (a) GOSIF, (b) GLASS, (c) PML, (d) TL and CSM using correlation-difference metric with Kendall's rank correlation (Corr) between detrended anomaly soil moisture (SM) and evapotranspiration (ET) from (e) CAMELE, (f) GLASS, (g) PML, and (h) SEBAL and Corr between detrended anomaly VPD and those ET products. And (i–w) the basin-average values of ZGE: Zhungaer Basin, PR: Pearl River Basin, YTR: Yangtze River Basin, SWR: Southwestern River Basin, TR: Tarim Basin, SR: Songhua River Basin, CT: Changthang Region, NM: Inner Mongolian Plateau Region, LR: Liaohe River Basin, YR: Yellow River Basin, HR: Huaihe River Basin, HC: Hexi Corridor Region, HAR: Haihe River Basin, SER: Southeastern River Basin, QB: Qaidam Basin.**

9. Figures 3 and 4: consider use noun as the subject in the titles rather than using verbs

**Response:** Thanks for your suggestion. We have changed the titles to

"Figure 3: Variations of soil moisture and evaporative fraction at (a) Xilingela in 2004, (c) Damshung in 2004, (e) CN-Sw2 in 2011, (g) CN-Du2 in 2007, (i) CN-Cng in 2010, (k) Miyun in 2009, (m) Huailai in 2015, (o) Qianyanzhou in 2010. Variations of covariance (referred to as Cov) between vapor pressure deficit and gross primary production, and correlation-difference metric (referred to as CorrVPD) at (b) Xilingela in 2004, (d) Damshung in 2004, (f) CN-Sw2 in 2011, (h) CN-Du2 in 2007, (j) CN-Cng in 2010, (l) Miyun in 2009, (n) Huailai in 2015, (p) Qianyanzhou in 2010."

and "Figure 4: Spatial pattern of number of wet binary bit at 10 cm depth using covariance between vapor pressure deficit (VPD) and gross primary production (GPP) from (a) GOSIF, (b) GLASS, (c) PML, (d) TL. Spatial pattern of number of wet binary bit at 10 cm depth using correlation-difference metric with Kendall's rank correlation (Corr) between detrended anomaly of soil moisture (SM) and evapotranspiration (ET) from (e) CAMELE, (f) GLASS, (g) PML, and (h) SEBAL and Corr between detrended anomaly of VPD and those ET products. (i) Disparity of soil moisture regimes among all methods and (j) the percentage of area with p < 0.05. ZGE: Zhungaer Basin, PR: Pearl River Basin, YTR: Yangtze River Basin, SWR: Southwestern River Basin, TR: Tarim Basin, SR: Songhua River Basin, CT: Changthang Region, NM: Inner Mongolian Plateau Region, LR: Liaohe River Basin, YR: Yellow River

Basin, HR: Huaihe River Basin, HC: Hexi Corridor Region, HAR: Haihe River Basin, SER: Southeastern River Basin, QB: Qaidam Basin."

10. Line 199: "9-day moving windows"?

**Response:** We have changed the sentence to "Hence, we concentrated on the warm season, June–September, which includes 16 data each year with 9 covariance values within 8-value moving windows. CSM was conducted in each grid cell using satellite-based ET and GPP over the period 2001–2018."

---

## Author Response (AR2)

Dear authors,

Your manuscript has now been reviewed by one of the reviewers who also reviewed the manuscript previously. As you can see, the reviewer is happy with the revised version but has a number of minor issues that will need to be addressed. As a result, I am happy to conditionally accept your manuscript for final publication in HESS, provided your response to the remaining issues is sufficient. In order to make the process as efficient as possible, the manuscript will not be returned to the referee and I will do the evaluation myself. I am looking forward to receiving your revised version.

Best regards

Ryan Teuling

**Response:** Thank you for your timely letter concerning our manuscript (No. HESS-2024-105) on the first day of 2025. We tried our best to improve the manuscript and made some changes in the manuscript. These changes will not influence the content and framework of the paper. We appreciate the Editor and Reviewer's warm work earnestly and hope that the correction will be approved. Revised portions are marked in red on the paper. The main corrections in the paper and the responses to the reviewer's comments are as follows:

......................................................................

**Response to Reviewer #1:**

Reviewer #1: Thanks to the authors for patiently addressing my comments and suggestions and providing substantial additional analyses. I am satisfied with the modifications to the manuscript. I still have a few comments, but those are minor and I concur with closing the review cycle. — Hsin.

**Response:** Thank you very much for your invaluable comments and suggestions. In the new version, we mainly integrate the two paragraphs of the conclusion, increase the importance and future potential of the research results, and modify some details in the full paper.

1. The conclusion needs improvement. The second paragraph in the conclusion **largely repeats the description of results**. You have sentences like "... **is essential for** the vegetation state and corresponding land-atmosphere coupling" and "**offering valuable insights** into the potential water limitation on ecosystems under comparable SM circumstances" in the abstract. Therefore, your conclusion should **echo these statements** and **provide some prospects** for future relevant studies.

**Response:** We changed the conclusion as follows:

"Our main accomplishment is observing and identifying water and energy limit shifts using multi-source satellite-based water and carbon fluxes over China. These shifts show which areas are more likely to be affected by climate change. To do so, we first examined the consistency of ET and GPP derived from the site- and satellite-based grid observations and the consistency of CSM derived from the EF-SM, covariance, and correlation-difference methods. CSM detected by the covariance between VPD and GPP and CSM using the correlation-difference metric using VPD, ET, and SM matched well with CSM using the EF-SM method at the site scale, suggesting that these methods could detect large-scale CSM. According to satellite-based CSM from four ET products, four GPP products, and the latest SM dataset, surface water- and energy-limited regimes varied among land cover types, soil textures, and water resource subregions; soil textures of clay and land cover types of grassland had a large range of SM within water-limited regimes. Based on the spatial pattern of CSM, we further attributed the dominant

factor of Δcorr and discovered that VPD was the most important predictor across 24% of Pearl River Basin and 19% of Tarim Basin. However, unlike the declining VPD in Pearl River Basin, the increasing VPD aggravated the water stress in Tarim Basin, especially for the more fragile grassland in these areas. As environmental change and extreme disturbances affect CSM, future research directions will aim at the impact of hydraulic projects such as inter-basin water transfers on CSM, the impact of extreme disturbances such as tropical cyclones and wildfires on CSM, and possible changes in CSM.

This study used multi-source satellite-based water and carbon fluxes and different methods to detect CSM, and more efforts were put into the evaluation and validation of CSM. 18 years of datasets used for CSM were quite typical of the long-term climatology of continental wetness. Since CSM, an emerging property, is generated by multiple processes occurring on the land surface, in the atmosphere, and at the interface between them, uncertainties of ET and GPP from the algorithm, uncertainties of SM from ground sampling, and enhanced land-atmosphere coupling due to external forcing all contribute to CSM uncertainties. We emphasize that SM behavior below and above CSM determines ET and GPP and that water-limited regimes of the SM range depend on CSM. Water and carbon fluxes are vulnerable to the sensitivity of Δcorr to hydrological, meteorological, and ecological predictors. Accordingly, the water and carbon algorithm should consider water-energy limit shifts to improve the simulation accuracy. Thus, applying our new understanding of Δcorr and CSM under changing land-atmosphere conditions will provide a more complete perspective of the evolution of regional terrestrial ecosystems over extended periods."

2. I suggest using ΔCorrVPD instead of CorrVPD to emphasize that it represents a difference. This would also be consistent with the notation in Denissen et al., 2020.

**Response:** In the revised version, we used Δcorr same with Denissen et al., 2020, instead of CorrVPD.

3. Line 26: "higher CSM than SM, making them in water-limited regimes." What does SM refer to? Should it be mean SM?

**Response:** Yes, we changed it to "CSM for grassland and clay was higher than average SM, making them in water-limited regimes".

4. Line 40: "relationship between SM and leaf conductance follows a linear trend." I think "trend" here is not the best wording, as it has a specific meaning in climate studies. Also, please indicate whether the relationship is positive or negative.

**Response:** We changed it to "SM and leaf conductance follow a positive linear relationship".

5. Line 174: "CMS" should be "CSM."?

**Response:** We changed it to "CSM".